# Mitigating Confirmation Bias in Semi-supervised Learning via Efficient Bayesian Model Averaging

**Charlotte Loh**                                                    *cloh@mit.edu*
*MIT EECS*
*MIT-IBM Watson AI Lab*

**Rumen Dangovski**                                          *rumenrd@mit.edu*
*MIT EECS*

**Shivchander Sudalairaj**                                   *shiv.sr@ibm.com*
*MIT-IBM Watson AI Lab*

**Seungwook Han**                                              *swhan@ibm.com*
*MIT EECS*
*MIT-IBM Watson AI Lab*

**Ligong Han**                                            *hanligong@gmail.com*
*Rutgers University*
*MIT-IBM Watson AI Lab*

**Leonid Karlinsky**                                         *leonidka@ibm.com*
*MIT-IBM Watson AI Lab*

**Marin Soljačić**                                           *soljacic@mit.edu*
*MIT Physics*

**Akash Srivastava**                               *akash.srivastava@ibm.com*
*MIT-IBM Watson AI Lab*

**Reviewed on OpenReview:** *https://openreview.net/forum?id=PRrKOaDQtQ&*

## Abstract

State-of-the-art (SOTA) semi-supervised learning (SSL) methods have been highly successful in leveraging a mix of labeled and unlabeled data, often via self-training or pseudo-labeling. During pseudo-labeling, the model's predictions on unlabeled data are used for training and may result in confirmation bias where the model reinforces its own mistakes. In this work, we show that SOTA SSL methods often suffer from confirmation bias and demonstrate that this is often a result of using a poorly calibrated classifier for pseudo labeling. We introduce BaM-SSL, an efficient Bayesian Model averaging technique that improves uncertainty quantification in SSL methods with limited computational or memory overhead. We demonstrate that BaM-SSL mitigates confirmation bias in SOTA SSL methods across standard vision benchmarks of CIFAR-10, CIFAR-100 and ImageNet, giving up to 16% improvement in test accuracy on the CIFAR-100 with 400 labels benchmark. Furthermore, we also demonstrate their effectiveness in additional realistic and challenging problems, such as class-imbalanced datasets and in photonics science.

# 1 Introduction

While deep learning has achieved unprecedented success in recent years, its reliance on vast amounts of labeled data remains a long standing challenge. Semi-supervised learning (SSL) aims to mitigate this by leveraging unlabeled samples in combination with a limited set of annotated data. In computer vision, two powerful techniques that have emerged are consistency regularization (Bachman et al., 2014; Sajjadi et al., 2016) and pseudo-labeling (also known as self-training) (Rosenberg et al., 2005; Xie et al., 2019b). Broadly, consistency regularization enforces that random perturbations of the unlabeled inputs produce similar predictions, while pseudo-labeling assigns artificial labels to unlabeled samples, which are then used to train the model. These two techniques are typically combined by minimizing the cross-entropy between pseudo-labels and predictions that are derived from differently augmented inputs, and have led to strong performances on vision benchmarks (Sohn et al., 2020; Assran et al., 2021).

In many SOTA SSL methods, a selection metric (Lee, 2013; Sohn et al., 2020) based on the model's confidence is often used in conjunction with pseudo-labeling, where only confident pseudo-labels are selected to update the model. As such, there is a need for proper confidence estimates; in other words, the calibration of the model should be of paramount importance. Model calibration (Guo et al., 2017) can be understood as a measure of how a model's output truthfully quantifies its predictive uncertainty, i.e. it denotes the alignment between its prediction confidence and its ground-truth accuracy. Apart from the importance of calibration arising from the selection metric, the use of cross-entropy minimization objectives common in SSL implies that models will naturally be driven to output high-confidence predictions (Grandvalet & Bengio, 2004). Having high-confidence predictions is highly desirable in SSL since we want the decision boundary to lie in low-density regions of the data manifold, i.e. away from labeled data points (Murphy, 2022). However, without proper calibration, a model would easily become over-confident. This is highly detrimental as the model would be encouraged to reinforce its mistakes, resulting in the phenomenon commonly known as *confirmation bias* (Arazo et al., 2019).

In this work, we propose to mitigate confirmation bias in semi-supervised learning by incorporating approximate Bayesian techniques, which have been widely known to improve uncertainty estimates (Wilson & Izmailov, 2020). Our main approach, BaM-, performs Bayesian Model averaging during pseudo-labeling by incorporating a Bayesian last layer to the model and is illustrated in Fig. 1. Broadly, BaM- can be described as follows; instead of single-fixed-value weights, the last layer are parameterized as Gaussian random variables and two main modifications are made during pseudo-labeling: 1) multiple weight samples are drawn from the layer and averaged to derive the predictions and 2) the selection criterion is based on their posterior variances (further details follow in Section 4.1). We further contextualize the novelty of BaM- against prior art in the SSL literature in Table 1. In contrast to prior methods, BaM- is designed to specifically target improving model calibration in order to mitigate confirmation bias.

Our contributions are summarized as follows:

1. We introduce BaM-, which is designed to mitigate confirmation bias via Bayesian model averaging in SOTA SSL methods based upon a selection metric. BaM- incorporates two new features to improve uncertainty estimation: 1) bayesian averaging over multiple weight samples and 2) a selection metric based on the variance of the predictions

2. We empirically demonstrate that BaM- effectively improves model calibration, resulting in better performances on standard benchmarks like CIFAR-10 and CIFAR-100, notably giving up to 16% gains in test accuracy.

3. We also demonstrate the efficacy of BaM- in more challenging and realistic scenarios, such as class-imbalanced datasets and a real-world application in photonic science.

# 2 Related Work

**Semi-supervised learning (SSL) and confirmation bias.** A fundamental problem in SSL methods based on pseudo-labeling (Rosenberg et al., 2005) is that of confirmation bias (Tarvainen & Valpola, 2017;

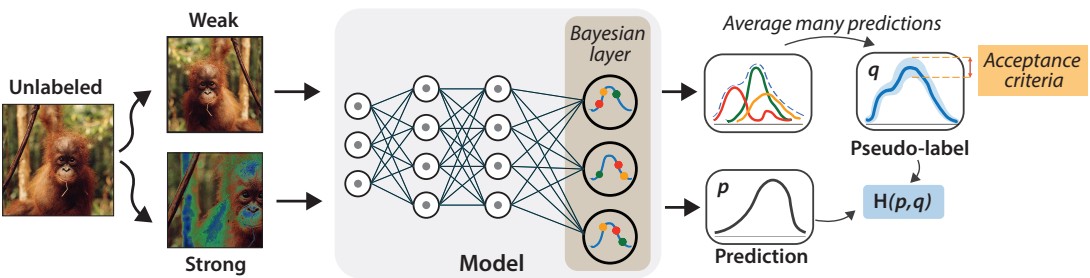

Figure 1: **Illustration of our Bayesian Model averaging (BaM) approach.** The last layer of the model has weights that are represented by probability distributions rather than the typical single fixed value. BaM- incorporates two main modifications to a traditional semi-supervised learning approach: 1) During pseudo-labeling of each unlabeled data point, multiple weights are sampled and averaged to derive the prediction; and 2) the selection criteria is based upon the variance over predictions.

Table 1: **Comparison of techniques used in BaM- with techniques used in prior art of SSL.**

| SSL method | Augmentation | Pseudo-label | Selection metric |
|---|---|---|---|
| Temporal ensemble (Laine & Aila, 2016) | Weak | Model from earlier step | - |
| Mean teacher (Tarvainen & Valpola, 2017) | Weak | EMA | - |
| UDA (Xie et al., 2019a) | Weak & strong | Sharpen | Logit thresholding |
| MixMatch (Berthelot et al., 2019) | Weak | Averaging aug + sharpen | - |
| FixMatch (Sohn et al., 2020) | Weak & strong | Hard labels | Logit thresholding |
| FlexMatch (Zhang et al., 2021) | Weak & strong | Hard labels | Class-wise logit threshold |
| Our main method BaM- | Weak & strong | **Posterior sampling** + sharpen | **Variance thresholding** |

Murphy, 2022), i.e. the phenomenon where a model overfits to incorrect pseudo-labels. Several strategies have emerged to tackle this problem; Guo et al. (2020) and Ren et al. (2020) looked into weighting unlabeled samples, Thulasidasan et al. (2019) and Arazo et al. (2019) proposes to use augmentation strategies like MixUp (Zhang et al., 2017), while Cascante-Bonilla et al. (2020) proposes to re-initialize the model before every iteration to overcome confirmation bias. Another popular technique is to impose a selection metric (Yarowsky, 1995) to retain only the highest quality pseudo-labels, commonly realized via a fixed threshold on the maximum class probability (Xie et al., 2019a; Sohn et al., 2020). Recent works have further extended such selection metrics to be based on dynamic thresholds, either in time (Xu et al., 2021) or class-wise (Zou et al., 2018; Zhang et al., 2021). Different from the above approaches, our work proposes to overcome confirmation bias in SSL by directly improving the calibration of the model through approximate Bayesian techniques.

**Model calibration and uncertainty quantification.** Proper estimation of a network's prediction uncertainty is of practical importance (Amodei et al., 2016) and has been widely studied. A common approach to improve uncertainty estimates is via Bayesian marginalization (Wilson & Izmailov, 2020), i.e. by weighting solutions by their posterior probabilities. Since exact Bayesian inference is computationally intractable for neural networks, a series of approximate Bayesian methods have emerged, such as variational methods (Graves, 2011; Blundell et al., 2015; Kingma et al., 2015), Hamiltonian methods (Springenberg et al., 2016) and Langevin diffusion methods (Welling & Teh, 2011). Other methods to achieve Bayesian marginalization also exist, such as deep ensembles (Lakshminarayanan et al., 2016) and efficient versions of them (Wen et al., 2020; Gal & Ghahramani, 2015), which have been empirically shown to improve uncertainty quantification. The concept of uncertainty and calibration are inherently related, where calibration is commonly interpreted as the frequentist notion of uncertainty. It is known that a well specified Bayesian model (i.e. one where the prior captures the model uncertainty) has a well-calibrated posterior (Gelman et al., 2021). Motivated by this, in our work we adopt some approximate bayesian techniques specifically for the context of semi-supervised learning in order to improve model calibration during pseudo-labeling and empirically validate their effectiveness. While other methods for improving model calibration exists (Platt, 1999; Zadrozny & Elkan, 2002; Guo et al., 2017), these are most commonly achieved in a post-hoc manner using a held-out validation set; instead,

we seek to improve calibration during training and with a scarce set of labels. In the intersection of SSL and calibration, Rizve et al. (2021) proposes to leverage uncertainty to select a better calibrated subset of pseudo-labels. Our work builds on a similar motivation, however, in addition to improving the selection metric with uncertainty estimates, we show that directly incorporating approximate Bayesian techniques into SSL methods can indeed improve calibration via its better-calibrated approximate posterior. Finally, our work also bears some close resemblance to acquistion functions used in Bayesian active learning. There, we seek data points for which the parameters under the posterior disagree about the outcome the most (Houlsby et al., 2011; Kirsch et al., 2019). In contrast, in semi-supervised learning where the goal is to reduce confirmation bias by selecting samples where the network is most certain about predicting, we instead seek data points for which the parameters under the posterior agree the most.

## 3 Notation and Background

Given a small amount of labeled data $\mathcal{L} = \{(x_l, y_l)\}_{l=1}^{N_l}$ (here, $y_l \in \{0,1\}^K$, are one-hot labels) and a large amount of unlabeled data $\mathcal{U} = \{x_u\}_{u=1}^{N_u}$, i.e. $N_u \gg N_l$, in SSL, we seek to perform a $K$-class classification task. Let $f(\cdot, \theta_f)$ be a backbone encoder (e.g. ResNet or WideResNet) with trainable parameters $\theta_f$, likewise let $h(\cdot, \theta_h)$ be a linear classification head, and $H$ denote the standard cross-entropy loss.

**SSL methods based on a selection metric.** Many SSL methods such as Pseudo-Labels (Lee, 2013), UDA (Xie et al., 2019a) and FixMatch (Sohn et al., 2020) use a selection metric in conjunction with pseudo-labeling to achieve SOTA performance. These methods minimizes a cross-entropy loss on augmented copies of unlabeled samples whose confidence exceeds a pre-defined threshold. Let $\alpha_1$ and $\alpha_2$ denote two augmentation transformations and their corresponding network predictions for sample $x$ to be $q_1 = h \circ f(\alpha_1(x))$ and $q_2 = h \circ f(\alpha_2(x))$, the total loss on a batch of unlabeled data has the following form:

$$L_u = \frac{1}{\mu B} \sum_{u=1}^{\mu B} \mathbb{1}(\max(q_{1,u}) \geq \tau) H(\rho_t(q_{1,u}), q_{2,u}) \tag{1}$$

where $B$ denotes the batch-size of labeled examples, $\mu$ a scaling hyperparameter for the unlabeled batch-size, $\tau \in [0,1]$ is a threshold parameter often set close to 1. $\rho_t$ is either a sharpening operation on the pseudo-labels, i.e. $[\rho_t(q)]_k := [q]_k^{1/t} / \sum_{c=1}^K [q]_c^{1/t}$ when soft-pseudo-labels are used, or a one-hot operation (i.e. $t \to 0$) when hard pseudo-labels are used. In the latter, we only care about the class where the maximum logit occurs. $\rho_t$ also implicitly includes a "stop-gradient" operation, i.e. gradients are not back-propagated from predictions of pseudo-labels. $L_u$ is combined with the expected cross-entropy loss on labeled examples, $L_l = \frac{1}{B} \sum_{l=1}^B H(y_l, q_{1,l})$ to form the combined loss $L_l + \lambda L_u$, with a scaling hyperparameter $\lambda$. Differences between Pseudo-Labels, UDA and FixMatch are detailed in Appendix D.1.

**Calibration metrics.** A popular empirical metric to measure a model's calibration is via the *Expected Calibration Error (*ECE*)*. Following (Guo et al., 2017; Minderer et al., 2021), we focus on a slightly weaker condition and consider only the model's most likely class-prediction, which can be computed as follows. Let $q_{\max}$ denote the model's confidence, or the prediction at the most likely class (i.e. the maximum logit value after the softmax), the model's confidence on a batch of $N$ samples are grouped into $M$ equal-interval bins, i.e. $\mathcal{B}_m$ contains the set of samples with $q_{\max} \in (\frac{m-1}{M}, \frac{m}{M}]$. ECE is then computed as the expected difference between the accuracy and confidence of each bin over all $N$ samples:

$$\text{ECE} = \sum_{m=1}^M \frac{|\mathcal{B}_m|}{N} |\text{acc}(\mathcal{B}_m) - \text{conf}(\mathcal{B}_m)| \tag{2}$$

where $\text{acc}(\mathcal{B}_m) = (1/|\mathcal{B}_m|) \sum_{i \in \mathcal{B}_m} \mathbb{1}(\text{argmax}(q_i) = y_i)$ and $\text{conf}(\mathcal{B}_m) = (1/|\mathcal{B}_m|) \sum_{i \in \mathcal{B}_m} q_{\max,i}$ with $y_i$ the true label of sample $i$. In this work, we estimate ECE using $M = 10$ bins. We also caveat here that while ECE is not free from biases (Minderer et al., 2021), we chose ECE over alternatives (Brier, 1950; DeGroot & Fienberg, 1983) due to its simplicity and widespread adoption.

## 4 Mitigating Confirmation Bias in Semi-supervised learning

As we see from Eq. (1), the model's confidence (the maximum softmaxed logit value) is used to determine if the pseudo-label of a particular unlabeled data point is used to update the model; as such it is important for the model to have proper confidence estimates, i.e. to be well-calibrated. More recently, temperature scaling (Guo et al., 2017) or other similar methods (Platt, 1999; Zadrozny & Elkan, 2002) have been highly effective towards model calibration. However, such methods pose several challenges in the semi-supervised setting; 1) they operate post-hoc, i.e. *after* training is completed, while in SSL the model needs to be calibrated constantly *during* training to reduce confirmation bias of pseudo-labels; 2) these methods use a held-out labeled set (typically around 10% of the total dataset size (Guo et al., 2017)) to perform calibration, while common benchmarks of SSL typically have label percentages less than that (up to as little as <1%), making it challenging to create a held-out set with so little data to begin with.

### 4.1 Mitigating Confirmation Bias with Bayesian Model Averaging

Given the above limitations of post-hoc calibration methods, in this work we propose to incorporate approxmate Bayesian methods such as Bayesian Neural Networks (BNN) into existing SSL methods. Bayesian models, when well-specified (i.e. where the prior captures the model's uncertainty), are known to produce well-calibrated posteriors (Gelman et al., 2021) and approximate Bayesian techniques have been widely empirically shown to produce well-calibrated uncertainty estimates in deep neural networks (Wilson & Izmailov, 2020; Lakshminarayanan et al., 2016; Blundell et al., 2015).

**Implementing a last-layer BNN.** In order to minimize the computational overhead and reduce the risk of overall poorer model accuracy arising from a full Bayesian approach Wenzel et al. (2020), we propose to only replace the **final layer** of the network, i.e. the linear classification head $h$, with a BNN layer. While there may be several options towards the implementation of the BNN layer, we propose to use a BNN with a variational posterior trained via stochastic variational inference (SVI) for computational efficiency. This would allow one to optimize both the non-bayesian backbone and the BNN layer simultaneously in a single backward pass, as opposed to other Bayesian approaches such as Hamiltonian Monte Carlo (Neal, 2012) which may require separate optimization loops.

**Stochastic variational inference in BaM-.** For notation convenience, we denote the input embedding to the BNN layer to be $v$ in this section. In SVI, we first assume a prior distribution on weights $p(\theta_h)$. Given some training data $\mathcal{D}_\mathcal{X} := (X, Y)$, we seek to calculate the posterior distribution of weights, $p(\theta_h|\mathcal{D}_\mathcal{X})$, which can then be used to derive the posterior predictive $p(y|v, \mathcal{D}_\mathcal{X}) = \int p(y|v, \theta_h)p(\theta_h|\mathcal{D}_\mathcal{X})d\theta_h$. This process is also known as "Bayesian model averaging" or "Bayesian marginalization" (Wilson & Izmailov, 2020). Since exact Bayesian inference is computationally intractable for neural networks, we adopt a variational approach following Blundell et al. (2015), where we learn a Gaussian variational approximation to the posterior $q_\phi(\theta_h|\phi)$, parameterized by $\phi$, by maximizing the evidence lower-bound (ELBO) (see Appendix C.1 for details). The $\text{ELBO} = \mathbb{E}_q \log p(Y|X; \theta) - KL(q(\theta|\phi)\|p(\theta))$ consists of a log-likelihood (data-dependent) term and a KL (prior-dependent) term. We provide some preliminary theoretical connections to generalization bounds via Corollary 1 in Appendix A. Corollary 1 shows that the generalization error is upper bounded by the negative ELBO, i.e. by maximizing the ELBO we may improve generalization. Apart from the last layer, the rest of the network is non-bayesian and are point values which are trained via regular Maximum Likelihood Estimation (MLE).

**Pseudo-labeling via BaM-.** As depicted in Fig. 1, pseudo-labeling in BaM- proceeds in two-stages: 1) $M$ weights from the BNN layer are sampled and predictions are derived from the Monte Carlo estimated posterior predictive, i.e. $\hat{q} = (1/M) \sum_m^M h(v, \theta_h^{(m)})$, and 2) the selection criteria is based upon their variance, $\sigma_c^2 = (1/M) \sum_m^M (h(v, \theta_h^{(m)}) - \hat{q})^2$, at the predicted class $c = \text{argmax}_{c'} [\hat{q}]_{c'}$. This constitutes a more intuitive measure of model uncertainty compared to the maximum logit value commonly used in prior SSL methods which **does not have an uncertainty interpretation**. In section 7, we verify through ablations that the variance of predictions is indeed more effective than the maximum logit value for mitigating confirmation bias. The variance is also highly intuitive — if the model's prediction has a large variance, it is highly uncertain

and the pseudo-label should not be accepted. We later show that better uncertainty estimates from BaM-effectively mitigates confirmation bias. In practice, as $\sigma_c^2$ decreases across training, we use a simple quantile $Q$ over the batch to define the threshold where pseudo-labels of samples with $\sigma_c^2 < Q$ are accepted, with $Q$ as a hyperparameter. Algorithm 1 shows a snippet of pseudo-code to highlight the main modifications introduced by BaM- during pseudo-labeling (a more complete version of the pseudo-code can be found in Appendix C.1).

We explore the effectiveness of BaM- by modifying upon SOTA SSL methods and denote them with the BaM-suffix, i.e. "BaM-X" incorporates approximate Bayesian Model averaging (BaM) during pseudo-labeling for SSL method X.

# 5 Experimental Setup

In all our experiments, we begin with and modify upon the original implementations of the baseline SSL methods. The backbone encoder $f$ is a Wide ResNet-28-2 and Wide ResNet-28-8 for the CIFAR-10 and CIFAR-100 benchmarks respectively. We use the default hyperparameters and dataset-specific settings (learning rates, batch size, optimizers and schedulers) recommended by the original authors for both the baselines and in BaM-. We set the weight priors in BaM- as unit Gaussians and use a separate Adam optimizer for the BNN layer with learning rate 0.01, no weight decay and impose the same cosine learning rate scheduler as the backbone. We set $Q = 0.75$ for the CIFAR-100 benchmark and $Q = 0.95$ for the CIFAR-10 benchmark; which are both linearly warmed-up from 0.1 in the first 10 epochs. As $Q$ is computed across batches, we improve stability by using a moving average of the last 50 quantiles.

**ECE and test accuracy evaluation.** In our experiments, we found that the test accuracy exhibits a considerable amount of noise across training, especially in label-scarce settings. Sohn et al. (2020) proposes to take the median accuracy of the last 20 checkpoints, while Zhang et al. (2021) argues that this fixed training budget approach is not suitable when convergence speeds of the algorithms are different (as the faster converging algorithm would over-fit more severely at the end) and thus report also the overall best accuracy. In our experiments, we adopt a balance between the two aforementioned approaches: we consider the median of 20 checkpoints around the best accuracy checkpoint as the *convergence criteria*, and report this value as the test accuracy. The ECE is reported when the model reaches this convergence criteria (one could also also aggregate the ECEs up till convergence — we found this gave similar trends and thus report the simpler metric).

---

**Algorithm 1** Snippet of PyTorch-style pseudocode showing pseudo-labeling in BaM-UDA.

```
# Q: quantile parameter
# num_samples: number of weight samples

q_list = []
for x_weak, x_strong in unlabeled_loader:
    z_weak, z_strong = encoder(x_weak), encoder(x_strong) # get representations
    mean_weak, std_weak = bayes_predict(bayes_classifier, z_weak) # get mean and std of predictions
    q_list.pop(0) if len(q_list) > 50 # keep 50 most recent quantiles
    q_list.append(quantile(std_weak, Q))
    accept_mask = std_weak.le(q_list.mean()) # determine acceptance for samples with small std

    # compute unlabeled loss using soft pseudo-labels on accepted samples
    loss_unlab = cross_entropy_loss(bayes_classifier(z_strong), sharpen(mean_weak)) * accept_mask
    loss_kl = KL_loss(bayes_classifier) # prior-dependent (data-independent) loss
    loss = loss_unlab + loss_kl

def bayes_predict(h, z):
    outputs = stack([h(z).softmax(-1) for _ in range(num_samples)]) # sample weights
    return outputs.mean(), outputs.std() # mean and std of predictions
```

---

Table 2: **BaM- in SSL** showing "Test accuracy (%) / ECE". BaM- improves calibration and result in better test accuracies, consistently across all benchmarks and across two SSL methods, FixMatch (FM) (Sohn et al., 2020) and UDA (Xie et al., 2019a). For each benchmark, results are averaged over 3 random dataset splits.

| | CIFAR-10 | | CIFAR-100 | | |
| --- | --- | --- | --- | --- | --- |
| | 250 labels | 2500 labels | 400 labels | 4000 labels | 10000 labels |
| FM (repro) | $95.0_{\pm 0.19}$ / $0.046_{\pm 0.002}$ | $95.7_{\pm 0.03}$ / $0.039_{\pm 0.0}$ | $56.4_{\pm 1.6}$ / $0.366_{\pm 0.017}$ | $74.2_{\pm 0.2}$ / $0.183_{\pm 0.003}$ | $78.1_{\pm 0.2}$ / $0.147_{\pm 0.001}$ |
| BaM-FM (ours) | $95.1_{\pm 0.07}$ / $0.044_{\pm 0.000}$ | $95.7_{\pm 0.1}$ / $0.039_{\pm 0.0}$ | $59.0_{\pm 1.4}$ (↑2.6) / $0.331_{\pm 0.015}$ | $74.8_{\pm 0.09}$ (↑0.6) / $0.171_{\pm 0.002}$ | $78.1_{\pm 0.2}$ / $0.139_{\pm 0.002}$ |
| UDA (repro) | $94.1_{\pm 0.6}$ / $0.053_{\pm 0.006}$ | $95.7_{\pm 0.05}$ / $0.039$ | $44.1_{\pm 0.7}$ / $0.473_{\pm 0.013}$ | $72.9_{\pm 0.01}$ / $0.189_{\pm 0.003}$ | $77.2_{\pm 0.3}$ / $0.154_{\pm 0.002}$ |
| BaM-UDA (ours) | $\mathbf{95.2}_{\pm 0.04}$ (↑1.1) / $\mathbf{0.042}_{\pm 0.00}$ | $\mathbf{95.9}_{\pm 0.08}$ (↑0.2) / $\mathbf{0.038}$ | $\mathbf{60.3}_{\pm 0.6}$ (↑16.2) / $\mathbf{0.314}_{\pm 0.005}$ | $\mathbf{75.2}_{\pm 0.1}$ (↑2.3) / $\mathbf{0.165}_{\pm 0.002}$ | $\mathbf{78.3}_{\pm 0.2}$ (↑1.1) / $\mathbf{0.138}_{\pm 0.003}$ |

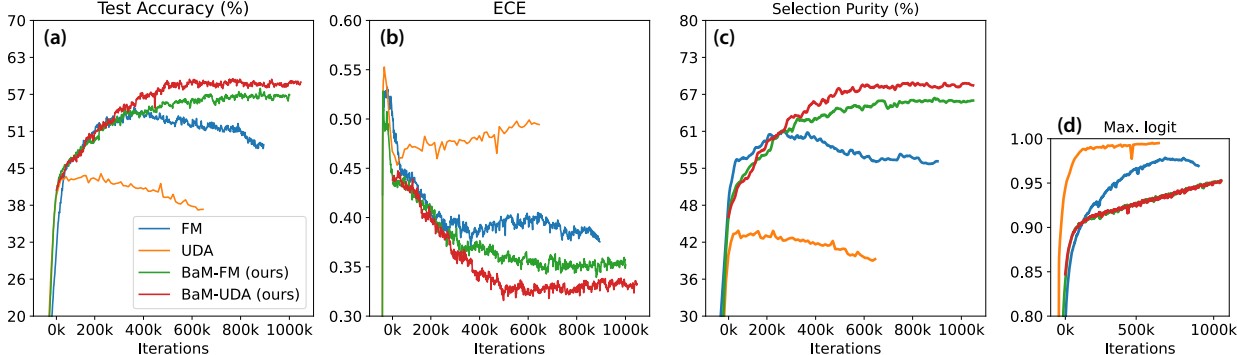

Figure 2: **FixMatch, UDA, BaM-FM and BaM-UDA across training** on CIFAR-100 with 400 labels (a) Test accuracies and (b) ECE as a function of training time. (c) Selection purity shows the accuracy of unlabeled samples that are accepted by the selection metric and (d) Max. logit shows the model's confidence, i.e. the average maximum class probability of all unlabeled samples.

# 6 Results

## 6.1 BaM- improves model calibration and consistently leads to better SSL performances

Results on the CIFAR-10 and CIFAR-100 benchmarks for various number of labels are depicted in Table 2. Table 2 demonstrates that BaM- successfully reduces the ECE over the baselines across all the benchmarks and as a result, we also attained significant improvements in test accuracies, notably up to 16.2% (for UDA on CIFAR-100-400 labels). Interestingly, while the baseline FixMatch outperforms UDA across all the benchmarks, improving calibration in UDA (i.e. BaM-UDA) allows it to outperfom both FixMatch and BaM-FixMatch. A key difference between FixMatch and UDA is the use of hard pseudo-labels in FixMatch (i.e. $t \to 0$ in $\rho_t$ defined in Section 3) versus soft pseudo-labels in UDA (with $t = 0.4$); this suggests that a Bayesian classifier is more effective in conjunction with soft pseudo-labels. Leveraging on this insight, we set $t = 0.9$ in BaM-UDA (also see Section 7.2 for ablations on $t$), resulting in consistently better performances over the two SSL baselines across all CIFAR benchmarks.

Overall, we found that the improvements in both calibration and test accuracy are more significant for label-scarce settings, as expected, since the problem of confirmation bias is more acute there and BaM- can provide greater benefits by mitigating this. We explore even more extreme settings in Appendix I, when there are only 1 or 2 labels per class in CIFAR-100. In the setting with only 1 label per class, BaM-UDA gives a decent 35.2% accuracy. In contrast UDA gives a poor 8.2% accuracy in such a challenging setting. Furthermore, the additional computation overhead incurred from BaM- is minimal, adding approximately only 2-5% in wall-clock time for the CIFAR-100 settings (see Appendix H). While previous works (Sohn et al., 2020) also include extreme low label settings such as CIFAR-10-40 labels, we found this benchmark to be highly sensitive to the random initialization and different splits of the data, giving up to 2% variance with the exact same SSL method and thus we exclude them in our study. Such variance in CIFAR-10 is significant, given that all methods are giving accuracies 95-96% and we do not realistically expect gains of 2% or more.

Table 3: **Long-tailed CIFAR-10 & CIFAR-100** showing "Test accuracy (%) / ECE". We use 10% of labels from each class. For better interpretation, we also show the supervised (100% labels) accuracy reported in Cao et al. (2019), which use a different architecture, ResNet-32, and an algorithm targeted for long-tailed problems.

| | CIFAR-10-LT | | CIFAR-100-LT | |
|---|---|---|---|---|
| | $\alpha = 10$ | $\alpha = 100$ | $\alpha = 10$ | $\alpha = 100$ |
| FixMatch | 91.3 / 0.073 | 70.0 / 0.26 | 48.8 / 0.38 | 28.6 / 0.55 |
| BaM-UDA (ours) | **91.6** (↑0.3) / **0.067** | **71.2** (↑1.2) / **0.24** | **53.6** (↑4.8) / **0.32** | **31.9** (↑3.3) / **0.50** |
| Supervised (reported) | 88.2 | 77.0 | 58.7 | 42.0 |

## 6.2 BaM- improves performances by reducing confirmation bias

To further understand how BaM- mitigates confirmation bias, we track the test accuracy, ECE, model confidence and ground-truth accuracy of *accepted* pseudo-labels (i.e. the "selection purity") over the course of training for the baselines and BaM-, as shown in Fig. 2. While the baselines learn effectively for the initial stages of training, learning is eventually hindered. Due to the entropy minization objective, the model is encouraged to output increasingly confident predictions (as evident from the growing model confidence in Fig. 2d). Thus, in the absence of explicit calibration, the baselines quickly become *over-confident*, resulting in confirmation bias where the model reinforces its mistakes. Confirmation bias in the baselines is particularly evident from the selection purity (i.e. the ground truth accuracy of accepted pseudo-labels) in Fig. 2c — after a short amount of training, the selection purity starts to drop suggesting that the model begins to accept pseudo-labels that it makes mistakes on. In contrast, BaM- successfully mitigates confirmation bias as evident from the constantly improving selection purity, thus promoting learning for longer periods to result in better final performances. Further ablation studies are discussed in Section 7.2 and Appendix G.

## 6.3 BaM- is more effective in class-imbalanced settings

**Long-tailed image datasets.** Datasets in the real world are often long-tailed or class-imbalanced, where some classes are more commonly observed while others are rare. We curate long-tailed versions from the CIFAR datasets following Cao et al. (2019), where $\alpha$ indicates the imbalance ratio (i.e. ratio between the sample sizes of the most frequent and least frequent classes). We randomly select 10% of the samples in each class to form the labeled set; see further details in Appendix E. We use the best performing baseline method from Table 2, i.e. FixMatch (FM), as our baseline and compare against BaM-UDA. Results are shown in Table 3, where BaM-UDA achieves consistent improvements over FM in both calibration and accuracy across all benchmarks. Notably, gains from BaM- are more significant than those in the class-balanced settings (for e.g. BaM-UDA improves upon FM by a smaller margin of 1.5% on the CIFAR-100-4000 labels benchmark which is also approximately 10%). Further analysis are provided in Appendix E.2, where test samples were separated into three groups depending on the number of samples per class and test accuracies are plotted for each group.

Table 4: **Photonic crystals (PhC) band gap prediction** showing "Test accuracy (%) / ECE". The fully-supervised (100% labels) accuracy is 88.5%.

| | PhC-10% | PhC-1% |
|---|---|---|
| FixMatch | 78.8 / 0.098 | 55.0 / 0.385 |
| BaM-UDA | **81.0** (↑2.2) / **0.052** | **56.9** (↑1.9) / **0.356** |

**Photonics science.** A practical example of a real-world domain where long-tailed datasets are prevalent is that of science – samples with the desired properties are often much rarer than trivial samples. Further, SSL is highly important in scientific domains since labeled data is particularly scarce (owing to the high resource cost needed for data collection). To demonstrate the effectiveness of BaM-, we adopt a problem in photonics (Loh et al., 2022), where the task is a 5-way classification of photonic crystals (PhCs) based on their band gap sizes. A brief summary and visualization of this dataset are available in Appendix F. We explored

an approach similar to FixMatch for the baseline and similar to BaM-UDA for ours (some modifications were needed in the augmentation strategies to respect the correct physics of this problem; see Appendix F). Results are shown in Table 4, where we demonstrate the consistency of BaM-UDA's effectiveness in improving calibration and accuracies in this real-world problem.

## 7    Ablation studies on BaM

### 7.1    Ablating the key components of BaM

BaM- consists of two main features; 1) several weight samples are taken from the BNN layer and averaged to derive the predictions, and 2) the selection criteria is modified to be based upon the variance of the samples. To further investigate the effect of each feature, we perform ablation studies on BaM-UDA to isolate the contribution of averaging predictions from the contribution of replacing the selection metric. Results are shown in Table 5, rows indicated with "BNN no $\sigma^2$" show experiments using a BNN layer in BaM-UDA *only for averaging predictions* while maintaining the original selection metric, i.e. pseudo-labels are accepted if the maximum prediction class probability is greater than $\tau = 0.95$. Comparing the first two rows, indeed we see that by not using the uncertainty estimates from the BNN, we already get some improvements (minor improvements in cases where labels are not so scarce). The difference between the last two rows show the effect of replacing the selection metric and indeed we observe larger and consistent gains across the benchmarks from doing so.

Table 5: **Uncertainty estimate by BNN.** Ablating the importance of uncertainty estimate provided by the variance of BNN predictions. "BNN no $\sigma^2$" indicates that the BNN layer is only used for bayesian model averaging, i.e. predictions are replaced by the posterior predictive but selection metric still follows the baseline, i.e. pseudolabels are accepted if maximum logit value after the softmax $> 0.95$. Cyan indicates the default configuration.

|  | CIFAR-100 | |
|---|---|---|
|  | 400 | 4000 |
| UDA (t=0.4) | 44.0 / 0.491 | 72.9 / 0.185 |
| BaM-UDA, BNN no $\sigma^2$ (t=0.4, $\tau$=0.95) | 48.3 / 0.418 | 73.0 / 0.184 |
| BaM-UDA, BNN no $\sigma^2$ (t=0.9, $\tau$=0.95) | 54.2 / 0.368 | 74.5 / 0.170 |
| BaM-UDA (t=0.9) | 59.7 / 0.327 | 75.3 / 0.167 |

### 7.2    Importance of sharpening temperature

From our results in Section 6, we found that BaM was more effective in conjunction with soft pseudo-labels. In Table 6, we show ablation experiments on the temperature of the sharpening operation on the CIFAR-100-400 labels benchmark. Overall, we observe a trend that softer pseudo-labels (i.e. reducing the sharpening of pseudo-labels) led to better calibration and improved test performance. As such, in our experiments we modify upon the original sharpening parameter of UDA and set $t = 0.9$ for BaM-UDA in all our benchmarks.

Table 6: Ablation of sharpening temperature in BaM-UDA. Dataset is CIFAR-100 with 400 labels. Highlighted in cyan is the main configuration used.

| $t$ | Test Accuracy | ECE |
|---|---|---|
| 0.4 | 57.9 | 0.344 |
| 0.8 | 58.1 | 0.340 |
| 0.9 | 59.7 | 0.327 |
| 1.0 | 59.2 | 0.334 |

## 8 Conclusion and Broader Impact

Since confirmation bias is a fundamental problem in SSL, in this work we showed that it is imperative for the model to have proper uncertainty estimates, or be well-calibrated, to mitigate this problem. In particular, we empirically demonstrated that approximate Bayesian techniques such as a last Bayesian layer or weight averaging approaches can be used to improve a model's uncertainty estimates which can result in better model performance across a variety of SSL methods. We further underscore their importance in more challenging real-world datasets. We hope that our findings can motivate future research directions to incorporate techniques targeted for optimizing calibration during the development of new SSL methods. Furthermore, while the primary goal of improving calibration is to mitigate confirmation bias during pseudo-labeling, an auxiliary benefit brought about by our approach is a better calibrated network, i.e. one that can better quantify its uncertainty, which is highly important for real-world applications. A potential limitation in our work lies in the use of ECE as a metric to measure calibration which, while commonly used across literature, are not free from flaws (Nixon et al., 2019). However, in our work, we empirically demonstrate that despite their flaws, the ECE metric still provides good correlations to measuring confirmation bias and test accuracy. We provide further discussion of the societal impact and ethical considerations of our work in Appendix J.

### Acknowledgments

We thank Hao Wang, Pulkit Agrawal, Tsui-Wei Weng, Rogerio Feris, Samuel Kim and Serena Khoo for fruitful conversations and feedback.

This work was sponsored in part by the United States Air Force Research Laboratory and the United States Air Force Artificial Intelligence Accelerator and was accomplished under Cooperative Agreement Number FA8750-19-2-1000. The views and conclusions contained in this document are those of the authors and should not be interpreted as representing the official policies, either expressed or implied, of the United States Air Force or the U.S. Government. The U.S. Government is authorized to reproduce and distribute reprints for Government purposes notwithstanding any copyright notation herein. This work was also sponsored in part by the the National Science Foundation under Cooperative Agreement PHY-2019786 (The NSF AI Institute for Artificial Intelligence and Fundamental Interactions, http://iaifi.org/) and in part by the Air Force Office of Scientific Research under the award number FA9550-21-1-0317.

C.L. also acknowledges financial support from the DSO National Laboratories, Singapore. L.D. is supported in part by the Defense Advanced Research Projects Agency (DARPA) under Contract No. FA8750-19-C-1001. Any opinions, findings and conclusions or recommendations expressed in this material are those of the author(s) and do not necessarily reflect the views of DARPA.

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

# A Theoretical Connections to Generalization Bounds

In the training of BaM-, we adopt a variational approach and maximize the evidence lower-bound (ELBO),

$$\text{ELBO} = \mathbb{E}_q \log p(Y|X;\theta) - KL(q(\theta|\phi)\|p(\theta))$$

with notation consistent with those used in Section 4.1 of the main text. We motivate this approach by using Corollary 1 below, derived from the PAC-Bayes framework (for the complete proof, see Appendix A.1 below). The statement shows the generalization error bounds on the variational posterior in the SSL setting and suggests that this generalization error is upper bounded by the negative ELBO. This motivates our approach, i.e. by maximizing the ELBO we improve generalization by minimizing the upper bound to the generalization error. The second term on the right side of the inequality characterises the SSL setting, and vanishes in the supervised setting.

**Corollary 1** *Let $\mathcal{D}$ be a data distribution where i.i.d. training samples are sampled, of which $N_l$ are labeled, $(x,y) \sim \mathcal{D}^{N_l}$ and $N_u$ are unlabeled, $(x,\hat{y}) \sim \mathcal{D}^{N_u}$ where $\hat{y}_i$ $(y_i)$ denotes the model-assigned pseudo-labels (true labels) for input $x_i$. For the negative log likelihood loss function $\ell$, assuming that $\ell$ is sub-Gaussian (see Germain et al. (2016)) with variance factor $s^2$, then with probability at least $1 - \delta$, the generalization error (denoted as $\mathcal{L}_{\mathcal{D}}^{\ell}$) of the variational posterior $q(\theta|\phi)$ is given by,*

$$\mathbb{E}_{q(\theta|\phi)}\mathcal{L}_{\mathcal{D}}^{\ell}(q) \leq \frac{1}{N}[-\text{ELBO}] - \mathbb{E}_{q(\theta|\phi)}\left[\frac{1}{N_u}\sum_{i=1}^{N_u}\log\frac{p(\hat{y}_i|x_i;\theta)}{p(y_i|x_i;\theta)}\right] + \frac{1}{N}\log\frac{1}{\delta} + \frac{s^2}{2} \qquad (3)$$

## A.1 Proof of Corollary 1

PAC-Bayesian theory aims to provide bounds on the generalization risk under the assumption that samples are i.i.d.. While PAC-Bayesian bounds typically apply to bounded losses, Germain et al. (2016) extends them to unbounded losses (which is necessary in our work since it uses the unbounded log-loss). Their theoretical result is reproduced below in Theorem 1, under assumptions for sub-Gaussian losses. These bounds commonly assume the supervised setting; in this proof, our goal is to extend them to the semi-supervised learning setting where in addition to labels from an empirical set, our loss is also mediated by pseudo-labels assigned by the model.

Let $(X,Y) \sim \mathcal{D}^N$ denote $N$ i.i.d. training samples from the data distribution $\mathcal{D}$, where $(X,Y) = \left\{\{(x_i,y_i)\}_{i=1}^{N_l}, \{(x_i,\hat{y}_i)\}_{i=1}^{N_u}\right\}$ represents the set of $N_l$ labeled input-output pairs and $N_u$ pairs of unlabeled input and their model-assigned pseudo-labels $\hat{y}$. Here, $N_l + N_u = N$. Let the loss function be $\ell(f,x,y) \to \mathbb{R}$, the generalization risk on distribution $\mathcal{D}$, i.e. $\mathcal{L}_{\mathcal{D}}^{\ell}(f)$, and the empirical risk on the training set, i.e. $\mathcal{L}_{X,Y}^{\ell}(f)$, is given by:

$$\mathcal{L}_{\mathcal{D}}^{\ell}(f) = \mathbb{E}_{(x,y)\sim\mathcal{D}}\ell(f,x,y); \quad \mathcal{L}_{X,Y}^{\ell}(f) = \frac{1}{N_l}\sum_{i=1}^{N_l}\ell(f,x_i,y_i) + \frac{1}{N_u}\sum_{i=1}^{N_u}\ell(f,x_i,\hat{y}_i)$$

On assumptions that the loss is sub-Gaussian (see Boucheron et al. (2013) section 2.3), i.e. a loss function $\ell$ is sub-Gaussian with variance $s^2$ under a prior $\pi$ and $\mathcal{D}$ if it can be described by a sub-Gaussian random variable $v = \mathcal{L}_{\mathcal{D}}^{\ell}(f) - \ell(f,x,y)$, the generalization error bounds are given by Theorem 1 (Germain et al., 2016) below.

**Theorem 1** *(Corollary 4 from Germain et al. (2016)) Let $\pi$ be the prior distribution and $\hat{\rho}$ be the posterior. If the loss is sub-Gaussian with variance factor $s^2$, with probability at least $1 - \delta$ over the choice of $(X,Y) \sim \mathcal{D}^N$,*

$$\mathbb{E}_{f\sim\hat{\rho}}\mathcal{L}_{\mathcal{D}}^{\ell}(f) \leq \mathbb{E}_{f\sim\hat{\rho}}\mathcal{L}_{X,Y}^{\ell}(f) + \frac{1}{N}\left(KL(\hat{\rho}\|\pi) + \log(1/\delta)\right) + \frac{1}{2}s^2$$

Beginning from Theorem 1, the generalization error of the variational posterior $q(\theta|\phi)$ under the negative log likelihood loss and prior $p(\theta)$ in our SSL setting can be found by replacing $\hat{\rho}$ with $q(\theta|\phi)$ and $\ell$ with

$$\ell^{nll}(f, x_i, y_i) := -\log p(y_i|x_i; \theta);$$

$$\underset{f \sim q}{\mathbb{E}} \mathcal{L}^{\ell}_{\mathcal{D}}(f) \leq \underset{f \sim q}{\mathbb{E}} \mathcal{L}^{\ell}_{X,Y}(f) + \frac{1}{N} \left( KL(q(\theta|\phi)\|p(\theta)) + \log(1/\delta) \right) + \frac{1}{2}s^2$$

$$= \mathbb{E}_q \left[ \frac{1}{N_u} \sum_{i=1}^{N_u} -\log p(\hat{y}_i|x_i; \theta) + \frac{1}{N_l} \sum_{i=1}^{N_l} -\log p(y_i|x_i; \theta) \right]$$
$$+ \frac{1}{N} \left( KL(q(\theta|\phi)\|p(\theta)) + \log(1/\delta) \right) + \frac{1}{2}s^2$$

$$= \mathbb{E}_q \left[ \frac{1}{N_u} \sum_{i=1}^{N_u} -\log \frac{p(\hat{y}_i|x_i; \theta)}{p(y_i|x_i; \theta)} + \frac{1}{N_u} \sum_{i=1}^{N_u} -\log p(y_i|x_i; \theta) + \frac{1}{N_l} \sum_{i=1}^{N_l} -\log p(y_i|x_i; \theta) \right]$$
$$+ \frac{1}{N} \left( KL(q(\theta|\phi)\|p(\theta)) + \log(1/\delta) \right) + \frac{1}{2}s^2$$

$$= \mathbb{E}_q \left[ \frac{1}{N_u} \sum_{i=1}^{N_u} -\log \frac{p(\hat{y}_i|x_i; \theta)}{p(y_i|x_i; \theta)} + \frac{1}{N} \sum_{i=1}^{N} -\log p(y_i|x_i; \theta) \right] + \frac{1}{N} \left( KL(q(\theta|\phi)\|p(\theta)) + \log(1/\delta) \right) + \frac{1}{2}s^2$$

$$= \mathbb{E}_q \left[ -\frac{1}{N_u} \sum_{i=1}^{N_u} \log \frac{p(\hat{y}_i|x_i; \theta)}{p(y_i|x_i; \theta)} \right] + \mathbb{E}_q \left[ -\frac{1}{N} \log p(Y|X; \theta) \right] + \frac{1}{N} \left( KL(q(\theta|\phi)\|p(\theta)) + \log(1/\delta) \right) + \frac{1}{2}s^2$$

$$\underset{f \sim q}{\mathbb{E}} \mathcal{L}^{\ell}_{\mathcal{D}}(f) \leq \mathbb{E}_q \left[ -\frac{1}{N_u} \sum_{i=1}^{N_u} \log \frac{p(\hat{y}_i|x_i; \theta)}{p(y_i|x_i; \theta)} \right] + \frac{1}{N} \left[ -\text{ELBO} \right] + \frac{1}{N} \log(1/\delta) + \frac{1}{2}s^2$$

where we define $\text{ELBO} = \mathbb{E}_{q(\theta|\phi)} [\log p(Y|X; \theta)] - \text{KL}(q(\theta \,|\, \phi)\|P(\theta))$. In the later sections, Appendix C.1, we will see that this is the term we are maximizing in our loss objective. By maximizing the ELBO, i.e. minimizing the negative ELBO, we are minimizing the upper bound to the generalization error of our variational posterior. The first term in the last line adds a divergence measure between the pseudo-label prediction distribution and the ground truth distribution — in the fully supervised setting $\hat{y}_i \to y_i$ and this term vanishes.

## B    Exploring SSL methods without a selection metric

In the main text, we have looked at two popular SSL methods that are based on a selection metric. More recently, there is also a family of SSL methods based upon representation learning (Assran et al., 2021; Chen et al., 2020b; Caron et al., 2021), where the model is first trained to produce useful representations and subsequently fine-tuned on limited labeled data. This give rise to some interesting questions: Is confirmation bias still an issue in SSL methods without a selection metric? And if so, can BaM- or other approximate Bayesian techniques be used to mitigate confirmation bias? We explore these questions in this section, using PAWS (Assran et al., 2021) as a canonical example for the family of SSL methods based on representation learning approach. Details about the training loss used in PAWS are explained in Appendix B.1. The main difference PAWS has in contrast to the methods studied above is in its prediction generation technique — instead of a classification head, PAWS uses a non-parametric nearest neighbour classification method over its representations.

### B.1    SSL methods based on representation learning

**PAWS.**    More recently, there has been a newer family of SSL methods based upon visual representation learning Chen et al. (2020a;b); Assran et al. (2021). We use PAWS Assran et al. (2021) as a canonical example of a SSL method from this family. A key difference of PAWS as compared to other selection-metric based SSL methods is the lack of a parametric classfier layer. Instead, predictions are derived from a non-parametric soft-nearest-neigbour classifier based on representations. Let $z_1 = f(\alpha_1(x))$ and $z_2 = f(\alpha_2(x))$ be the representations for the two views from the backbone encoder, their pseudo-labels $(q_1, q_2)$ and the

unlabeled loss are given by:

$$q_i = \pi_d(z_i, \{z_s\}) = \sum_{s=1}^{B} \frac{d(z_i, z_s) \cdot y_s}{\sum_{s'=1}^{B} d(z_i, z_{s'})}; \quad L_\mathrm{u} = \frac{1}{2\mu B} \sum_{u=1}^{\mu B} H(\rho_t(q_{1,u}), q_{2,u}) + H(\rho_t(q_{2,u}), q_{1,u}) \qquad (4)$$

where $\{z_s\}_{s=1}^{B}$ is the set of representations of the labeled examples, $d(a,b) = \exp(a \cdot b/(\|a\|\|b\|\tau_p))$ is a similarity metric with temperature hyperparameter $\tau_p$ and all other symbols have the same meanings defined before in Section 3 of the main text. The combined training loss for PAWS is $L_u + L_\mathrm{me\text{-}max}$ where the latter is a regularization term $L_\mathrm{me\text{-}max} = H(\bar{q})$ that seeks to maximize the entropy of the average of predictions $\bar{q} := (1/(2\mu B)) \sum_{u=1}^{\mu B} (\rho_t(q_{1,u}) + \rho_t(q_{2,u}))$.

## B.2 Exploring other approximate Bayesian techniques

Since BaM- replaces the classification head and PAWS does not have one, we incorporate BaM- into PAWS by using a Bayesian last layer in the encoder and "Bayesian marginalizing" over the representations (also see Appendix B.4.1 for detailed formulation). Nonetheless, apart from Bayesian marginalization, BaM- is originally desirable towards improving uncertainty estimates in the selection metric. Since PAWS does not use a selection metric, instead of Bayesian marginalizing or aggregating over just one layer, one could seek to aggregate over the entire network. This can be most commonly acheived by approximate bayesian techniques such as model ensembling approaches Lakshminarayanan et al. (2016), which has been highly successful at improving uncertainty estimation. However, training multiple networks would add immense computational overhead. To perform this aggregation computational efficiently, we instead explored weight averaging approaches, such as Stochastic Weight Averaging (Izmailov et al., 2018) (SWA) and Exponential Moving Averaging (EMA) (Tarvainen & Valpola, 2017; He et al., 2020; Grill et al., 2020) and studied their role in mitigating confirmation bias during pseudo-labeling. Weight averaging however differs from traditional bayesian model averaging approaches typically used in ensembles since we are averaging in the weight-space (i.e. averaging model weights) instead of function-space (i.e. averaging predictions). The approximation of SWA to Fast Geometric Ensembles (Garipov et al., 2018) has been shown by previous works (Izmailov et al., 2018); however, to the best of our knowledge the connection of EMA to ensembling has not been formally shown.

**Weight averaging during pseudo-labeling** We maintain a separate set of non-trainable weights $\theta_g$ containing the aggregated weight average of $\theta_h$ which is used to produce pseudo-labels throughout training. In SWA, we update them via $\theta_g \leftarrow (n_a \theta_g + \theta_f)/(n_a + 1)$ at every iteration, where $n_a$ represents the total number of models in the aggregate and $\theta_f$ are the trainable parameters of our backbone encoder as before. In practice, we switch on SWA only after some amount of training. In EMA, this update is $\theta_g \leftarrow \gamma \theta_g + (1 - \gamma)\theta_f$, where $\gamma$ is a momentum hyperparameter controlling how much memory $\theta_g$ should retain at each iteration (see Appendix C.2 for pseudocode). While EMA has been previously explored in the context of SSL (Tarvainen & Valpola, 2017), to the best of our knowledge SWA has not been explored in SSL and more importantly, their link to mitigating confirmation bias in pseudo-labeling has not been explicitly shown.

Table 7: "Test accuracy (%) / ECE" for PAWS (Assran et al., 2021) and our methods. 'BaM-' refers to Bayesian Model averaging via a BNN final layer, while +SWA and +EMA are using weight averaging techniques. For each benchmark, results are averaged over 3 random dataset splits.

| | CIFAR-10 | CIFAR-100 | |
| --- | --- | --- | --- |
| | 2500 labels | 4000 labels | 10000 labels |
| PAWS (repro) | 95.3±0.1 / 0.046±0.001 | 71.5±0.3 / 0.232±0.005 | 75.6±0.1 / 0.198±0.002 |
| BaM-PAWS | 95.4±0.1 (↑0.1) / 0.046±0.001 | 72.5±0.3 (↑1.0) / 0.228±0.003 | 76.5±0.2 (↑0.9) / 0.195±0.002 |
| PAWS+SWA | 95.6±0.1 (↑0.3) / 0.046±0.001 | **74.5**±0.3 (↑3.0) / **0.193**±0.002 | **78.4**±0.4 (↑2.8) / **0.164**±0.003 |
| PAWS+EMA | **95.8**±0.0 (↑0.5) / **0.042**±0.000 | 73.5±0.2 (↑2.0) / **0.193**±0.005 | 77.2±0.1 (↑1.6) / 0.168±0.003 |

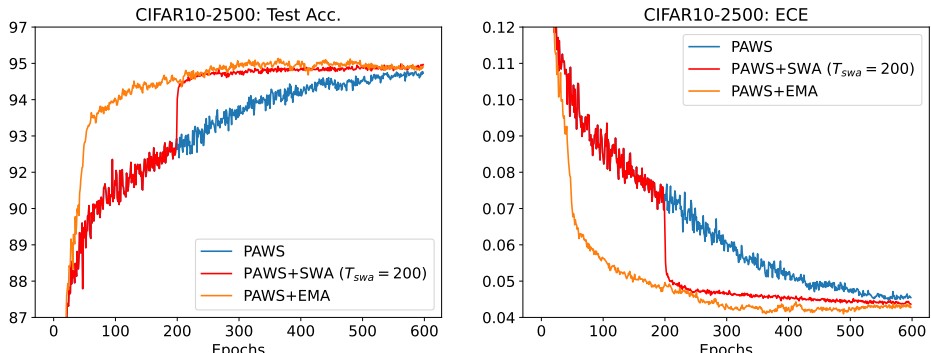

Figure 3: **Weight averaging techniques on PAWS for CIFAR-10 with 2500 labels** displays highly consistent relationships between increasing test accuracy and decreasing ECE.

### B.3 BaM- is useful, but weight averaging is more effective when a selection metric is absent

Results are shown in Table 7; while BaM- gave some improvements over the baselines across all benchmarks, the improvements from weight averaging approaches are consistently larger. This is unsurprising, since BaM- relies on stochasticity only in the last layer while weight averaging aggregates over the entire model. Since PAWS does not use a selection metric and simply "accepts all unlabeled samples", the "selection purity" is absent and thus we investigate the issue of confirmation bias by tracking the accuracy and model calibration on a held-out set as shown in Fig. 3. Incorporating weight averaging results in better accuracies on the held-out set compared to the baseline at every stage of training. This suggests that even in the absence of a selection metric, proper uncertainty estimates (or improved model calibration) can lead to improved SSL performance. The effectiveness of improved uncertainty estimates is particularly evident through PAWS+SWA—when SWA was switched on (at $T_{\mathrm{swa}} = 200$ epochs), model calibration was immediately improved (i.e. the ECE quickly dips) and the accuracy also correspondingly spikes. Furthermore, we also observe a significant improvement in convergence rates resulting from the weight averaging techniques, i.e. better accuracies can be attained in just a third of the number of training iterations of the baseline. Ablations on $T_{\mathrm{swa}}$ and momentum schedules are presented in **??**.

**Effective also in large-scale datasets like ImageNet.** A strength of representation learning SSL methods is in large-scale datasets like ImageNet, where PAWS outperforms selection metric based SSL methods (i.e. 73.9% vs 71.5% on the best performing selection metric based method, FixMatch for the ImageNet-10% benchmark). In this work, we use the 10% labels setting of Assran et al. (2021) with slight modifications to the default setting in order to fit on our system (see Appendix D.2 for implementation details) and explored weight averaging techniques SWA and EMA, given their effectiveness over BaM- in PAWS. Results are shown in Table 8, where our methods provide consistent improvements in calibration and test accuracy over the baseline.

Table 8: **ImageNet-10%** showing "Top-1 test accuracy (%) / ECE" for PAWS and our methods, trained for 200 epochs and fine-tuned with a linear head.

|  | IN-10% |
| --- | --- |
| PAWS | 74.4 / 0.186 |
| +SWA (ours) | 74.5 / 0.186 |
| +EMA (ours) | **75.1 / 0.183** |

### B.4 Further experimental details

#### B.4.1 Improving calibration in PAWS with BaM- (i.e. BaM-PAWS)

Because of the lack of a classifier layer, incorporating BaM- requires some modifications. Instead, we use a BaM- in the last layer in the encoder (which is a linear layer) and pseudo-labels are obtained by "Bayesian marginalizing" over the representations. More specifically, let $h$ be the Bayesian version of this layer, parameterized by $\theta_h$ and let $v$ be the input to this layer, pseudo-labels of BaM-PAWS (i.e. Eq. (4)) are

defined as:

$$q_i = \pi_d(\hat{z}, \{z_s\}) = \sum_{s=1}^{B} \frac{d(\hat{z}, z_s) \cdot y_s}{\sum_{s'=1}^{B} d(\hat{z}, z_{s'})}; \quad \hat{z} = (1/M) \sum_{m}^{M} h(v, \theta_h^{(m)}) \tag{5}$$

where $M$ is the number of samples taken from the Bayesian layer.

### B.4.2 Further training details for BaM-PAWS, PAWS+EMA, PAWS+SWA

**BaM-PAWS.** Unlike BaM- in selection-metric based SSL methods, we do not use a separate optimizer and simply impose a one-minus-cosine scheduler (i.e. scheduler (2) from the next paragraph) for the coefficient to the KL-divergence loss that goes from 0 to 1 in $T$ epochs where we simply picked the best from $T \in \{50, 100\}$.

**PAWS+SWA and PAWS+EMA.** We delay the onset of SWA to after some amount of training has elapsed (i.e. $T_{swa}$ epochs) since it is undesirable to include the randomly initialized weights to the aggregate. We set $T_{swa} = 200$ and $T_{swa} = 100$ for CIFAR-10 and CIFAR-100 respectively. After SWA was switched on, the weight aggregate was simply updated after every batch iteration. In contrast, EMA has a natural curriculum to "forget" older model parameters since more recent parameters are given more weight in the aggregate. We experimented with two schedules for $\gamma$: 1) using a linear warm-up from 0 to 0.996 in 50 epochs and then maintaining $\gamma$ at 0.996 for the rest of training and 2) using a one-minus-cosine scheduler starting from 0.05 and decreasing to 0 resulting in a 0.95 to 1 range for $\gamma$. We visualize these schedulers and include ablation studies on them and on $T_{\text{swa}}$ in **??**.

## C Formulation details and pseudocode of our methods

### C.1 Bayesian model averaging via a BNN final layer

Following the notations in Section 4.1, we denote the BNN layer to be $h$ and an input embedding to this layer to be $v$ in this section. We assume a prior distribution on weights $P(\theta_h)$ and seek to calculate the posterior distribution of weights given the empirical/training data, $P(\theta_h | \mathcal{D}_\mathcal{X})$, where $\mathcal{D}_\mathcal{X} := (X, Y)$, which can then be used to compute the posterior predictive during inference. As exact Bayesian inference is intractable for neural networks, we adopt a variational approach following (Blundell et al., 2015) to approximate the posterior with a Gaussian distribution parameterized by $\phi$, $q(\theta|\phi)$. From now, we will drop the $h$ index in $\theta_h$ for brevity. To learn the variational approximation, we seek to minimize the Kullback-Leibler (KL) divergence between the Gaussian variational approximation and the posterior:

$$
\begin{aligned}
\phi^* &= \text{argmin}_\phi \text{KL}\left(q(\theta|\phi) \, \| \, p(\theta|X,Y)\right) \\
&= \text{argmin}_\phi \int q(\theta|\phi) \log \frac{q(\theta|\phi)}{p(\theta|X,Y)} d\theta \\
&= \text{argmin}_\phi \mathbb{E}_{q(\theta|\phi)} \log \frac{q(\theta|\phi)p(Y|X)}{p(Y|X;\theta)p(\theta)} \\
&= \text{argmin}_\phi \mathbb{E}_{q(\theta|\phi)} \left[\log q(\theta|\phi) - \log p(Y|X;\theta) - \log p(\theta)\right] + \log p(Y|X) \\
&= \text{argmin}_\phi \text{KL}(q(\theta|\phi)\|p(\theta)) - \mathbb{E}_{q(\theta|\phi)} \log p(Y|X;\theta) + \log p(Y|X) \\
&= \text{argmin}_\phi \left([-\text{ELBO}] + \log p(Y|X)\right)
\end{aligned}
$$

where in the last line, $\text{ELBO} = \mathbb{E}_{q(\theta|\phi)}\left[\log p(Y|X;\theta)\right] - \text{KL}(q(\theta|\phi)\|p(\theta))$ is the evidence lower-bound which consists of a log-likelihood (data-dependent) term and a KL (prior-dependent) term. Since $\text{KL}\left(q(\theta|\phi) \, \| \, p(\theta|\mathcal{D}_\mathcal{X})\right)$ is intractable, we maximize the ELBO which is equivalent to minimizing the former up to the constant, $\log p(Y|X)$.

Each variational posterior parameter of the Gaussian distribution, $\phi$, consists of the mean ($\mu$) and the standard deviation (which is parametrized as $\sigma = \log(1 + \exp(\rho))$ so that $\sigma$ is always positive (Blundell et al., 2015)), i.e. $\phi = (\mu, \rho)$. To obtain a sample of the weights $\theta$, we use the reparametrization trick (Kingma et al., 2015) and sample $\epsilon \sim \mathcal{N}(0, I)$ to get $\theta = \mu + \log(1 + \exp(\rho)) \circ \epsilon$ where $\circ$ denotes elementwise multiplication.

**Algorithm 2** PyTorch-style pseudocode for Bayesian model averaging in UDA or FixMatch.

```
# f: backbone encoder network
# h: bayesian classifier
# KL_loss: KL term in evidence lower-bound
# H: cross-entropy loss
# Q: quantile parameter
# num_samples: number of weight samples
# method: 'UDA' or 'FM'
# shp: sharpen operation

q_list = []
for (xl, labels), xu in zip(labeled_loader, unlabeled_loader):
    x_lab, x_uw, x_us = weak_augment(xl), weak_augment(xu), strong_augment(xu)
    z_lab, z_uw, z_us = f(x_lab), f(x_uw), f(x_us) # get representations
    mean_uw, std_uw = bayes_predict(h, z_uw)
    q_list.pop(0) if len(q_list) > 50 # keep 50 most recent quantiles
    q_list.append(quantile(std_uw, Q))
    mask = std_uw.le(q_list.mean())

    # compute losses
    loss_kl = KL_loss(h) # prior-dependent (data-independent) loss
    loss_lab = H(h(z_lab), labels)
    if method == 'UDA':
        loss_unlab = H(h(z_us), shp(mean_uw)) * mask # sharpened soft pseudo-labels
    elif method == 'FM':
        loss_unlab = H(h(z_us), mean_uw.argmax(-1)) * mask # hard pseudo-labels
    loss = loss_lab + loss_unlab + loss_kl
    loss.backward()
    optimizer.step()

def bayes_predict(h, z):
    outputs = stack([h(z).softmax(-1) for _ in range(num_samples)]) # sample weights
    return outputs.mean(), outputs.std() # mean and std of predictions
```

In other words, we double the number of learnable parameters in the layer compared to a non-Bayesian approach; however, this does not add a huge computational cost since only the last layer is Bayesian and the dense backbone remains non-Bayesian.

Algorithm 2 shows the PyTorch-style pseudo-code for BaM- in selection-metric based methods (here showing asymmetric augmentation applicable for UDA or FixMatch). The main modifications upon the baseline methods include 1) computation of an additional KL loss term (between two Gaussians, i.e. the variational approximation and the prior), 2) taking multiple samples of weights to derive predictions and 3) replacing the acceptance criteria from using maximum probability to using standard deviation of predictions.

### C.2 Algorithm for SWA & EMA in PAWS

Algorithm 3 shows the pseudocode for the implementation of PAWS+SWA and PAWS+EMA in PyTorch. For brevity, we leave out details of the multicrop strategy, mean entropy maximization regularization and soft nearest neighbour classifier formulation which are all replicated from the original implementation. We defer readers to the original paper (Assran et al., 2021) for these details.

---

**Algorithm 3** PyTorch-style pseudocode for PAWS-SWA and PAWS-EMA.

---

```
# f: backbone encoder network
# g: weight aggregated encoder network
# shp: sharpen operation
# snn: PAWS soft nearest neighbour classifier
# H: cross entropy loss
# ME_max: mean entropy maximization regularization loss
# N: total number of epochs
# use_swa: boolean to use SWA
# use_ema: boolean to use EMA
# swa_epochs: number of epochs before switching on SWA
# gamma: momentum parameter for EMA

g.params = f.params # initialized as copy
g.params.requires_grad = False # remove gradient computations
num_swa = 0
for i in range(N):
    for x in loader:
        x1, x2 = augment(x), augment(x) # augmentations for x
        p1, p2 = snn(f(x1)), snn(f(x2))
        q1, q2 = snn(g(x1)), snn(g(x2))

        loss = H(p1, shp(q2))/2 + H(p2, shp(q1))/2 + ME_max(cat(q1,q2))
        loss.backward()
        optimizer.step()

        if use_swa:
            if i > swa_epochs: # update aggregate
                num_swa += 1
                g.params = (g.params * num_swa + f.params) / (num_swa + 1)
            else:
                g.params = f.params # weights are just copied
        elif use_ema:
            g.params = gamma * g.params + (1-gamma) * f.params # update momentum aggregate
        else:
            g_params = f.params # PAWS baseline
```

---

# D  Further implementation details

## D.1  Hyperparameters for various selection-metric based methods

All selection-metric based methods in this study uses an optimization loss function of the form of Eq. (1) from the main text, with differences in the hyperparameters $\mu$, $\lambda$, $\tau$, $\rho_t$ and $\alpha$. We use the hyperparameters from the original implementations. For Pseudo-Labels (Lee, 2013), $\mu = 1$, $\tau = 0.95$ and $\rho_{t=0}$ (i.e. hard labels); for UDA (Xie et al., 2019a), $\mu = 7$, $\tau = 0.8$, $\rho_{t=0.4}$ (i.e. soft pseudo-labels sharpened with temperature of 0.4); for FixMatch (Sohn et al., 2020), $\mu = 7$, $\tau = 0.95$, $\rho_{t=0}$ (i.e. hard labels). All methods use $\lambda = 1$. In addition, UDA and FixMatch uses asymmetric transforms for the two legs of sample and pseudo-label prediction, i.e. $\alpha_1$ is a weak transform (based on the standard flip-and-shift augmentation) and $\alpha_2$ is a strong transform (based on RandAugment (Cubuk et al., 2019)). Pseudo-Labels uses symmetric weak transforms for both legs.

In our calibrated versions of all these methods (i.e. "BaM-X"), we maintained the exact same hyperparameter configuration as its corresponding baseline. The only exception is the sharpening temperature of BaM-UDA, which uses $t = 0.9$ instead of $t = 0.4$ in UDA, as we found that calibration enables, and was highly effective with, the use of soft pseudo-labels).

## D.2  Implementation details for ImageNet experiments

We maintain the default 64 GPU training configuration recommended by the authors and make slight modifications to the default implementations to fit on our hardware. On our set up (which does not use Nvidia's Apex package due to installations issues), we found training to be unstable on half-precision and had to use full-precision training. In order to fit into memory, we had to decrease the number of images per class from 7 to 6, resulting in a slightly lower baseline performance from the reported for 200 epochs of training (see Table 4 in Assran et al. (2021) for the study on the correlation between the number of images per class and the final accuracy). On all ImageNet experiments on PAWS, we follow the validation and testing procedure of PAWS (Assran et al., 2021) and swept over the same set of hyperparameters during fine-tuning of the linear head. We report the ECE at the final checkpoint.

# E  Long-tailed CIFAR-10 and CIFAR-100

## E.1  Dataset preparation

We create long-tailed versions of CIFAR-10 and CIFAR-100 following the procedure from Cao et al. (2019), i.e. by removing the number of training examples per class from the standard training set with 50,000 samples. We create the class-imbalance unlabeled dataset with an exponential decay where the severity of the imbalance is given by the imbalance ratio $\alpha = \max_i(n_{u,i})/\min_i(n_{u,i}) \in \{10, 100\}$, where $n_{u,i}$ is the number of unlabeled examples for class $i$. The number of samples in the most frequent class is 5,000 for CIFAR-10 and 500 for CIFAR-100. To create the labeled set, we randomly select 10% of samples *from each class*, under the constrain that at least 1 sample for each class is included in the labeled set, i.e. $n_{l,i} = \min(1, 0.1 * n_{u,i})$, where $n_{l,i}$ is the number of labeled examples for class $i$. The total number of labeled and unlabeled examples for the different benchmarks in CIFAR-10-LT and CIFAR-100-LT are summarized in Table 9. The test set remains unchanged, i.e. we use the original (class-balanced) test set of the CIFAR datasets with 10,000 samples.

Table 9: **Dataset statistics for CIFAR-10-LT & CIFAR-100-LT** showing total number of examples in the labeled and unlabeled datasets for each benchmark.

|         | CIFAR-10-LT | | CIFAR-100-LT | |
|---------|:---:|:---:|:---:|:---:|
|         | $\alpha = 10$ | $\alpha = 100$ | $\alpha = 10$ | $\alpha = 100$ |
| Labeled | 2,041 | 1,236 | 1,911 | 1,051 |
| Total   | 20,431 | 12,406 | 19,573 | 10,847 |

We used the exact same configuration and hyperparameters as the original (non-long-tailed) CIFAR benchmarks, i.e. the architecture is WideResNet-28-2 for CIFAR-10-LT and WideResNet-28-10 for CIFAR-100-LT. All training hyperparameters used to obtain the results for FM and BaM-UDA in Table 3 of the main text also follow the ones from the original CIFAR benchmarks.

## E.2  Additional classwise results

Fig. 4 shows the test accuracies for the baseline (FM) and ours (BaM-UDA) after separating samples from the test set into three groups — the "Many" group which contains classes with more than 100 samples, the "medium" group which contains classes between 10 and 100 samples and the "few" group which contains classes less than 10 samples, for the benchmark of CIFAR-100-LT with $\alpha = 10$ and 10% labels. We see that BaM-UDA outperforms FM on every group; in addition, the gap between BaM-UDA and FM increases as the number of samples are more scarce, highlighting BaM-UDA's utility in improving accuracy over the baseline in the more difficult classes.

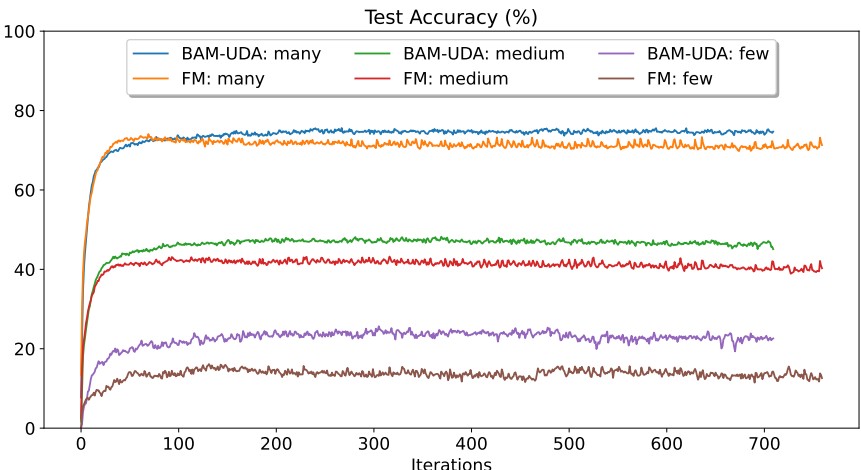

Figure 4: **CIFAR100-LT**. Accuracies for samples based on their their classwise sample frequency, "many" for classes with >100 samples, "medium" for classes between 10-100 samples and "few" for classes between <10 samples. Dataset is CIFAR-100-LT with $\alpha = 10$ and 10% labels.

## F  Semi-supervised Learning in Photonics Science

Semi-supervised learning is highly important and applicable to domains like Science where labeled data is particularly scarce, owing to the need for computationally expensive simulations and labor-intensive laboratory experiments for data collection. To demonstrate the effectiveness of our proposed techniques in the Science domain, we use an example problem in Photonics, adopting the datasets from Loh et al. (2022). The task we studied in this work is a 5-way classification of photonic crystals (PhC) based on their band gap sizes.

PhCs are periodically-structured materials engineered for wide ranging applications by manipulating light waves (Joannopoulos et al., 2008) and an important property of these crystals is the size of their band gap (often, engineers seek to design photonic crystals that host a substantial band gap (Christensen et al., 2020)). We adopt the dataset of PhCs from Loh et al. (2022), which consists of 32,000 PhC samples and their corresponding band gaps which had been pre-computed through numerical simulations (Johnson & Joannopoulos, 2001). Examples of PhCs and an illustration of band gap from the dataset is shown in Fig. 5. We binned all samples in the dataset into 5 classes based on their band gap sizes; since there was a preponderance (about 25,000) of samples without a band gap, we only selected 5,000 of them in order to limit the severity of class imbalance and form a new dataset with just 11,375 samples (see Fig. 5c). Notably the long-tailed distribution of this dataset is characteristic of many problems in science (and other real-world datasets), where samples with the desired properties (larger band gap) are much rarer than trivial samples. From this reduced dataset, we created two PhC benchmarks with 10% labels (PhC-10%) and 1% labels (PhC-1%), where 10% and 1% of samples *from each class* are randomly selected to form the labeled set respectively. The class-balanced test set is fixed with 300 samples per class (total of 1,500 samples), the unlabeled set consists of 9,876 samples and the labeled sets consists of 1,136 samples and 113 samples for PhC-10% and PhC-1% respectively.

For these benchmarks, we used a WideResNet-28-2 architecture, with a single channel for the first CNN layer, and made the following changes upon FixMatch and BaM-UDA. We set $\mu = 1$ and instead of $2^{20}$ iterations, we trained for 300 epochs (where each epoch is defined as iterating through the unlabeled set once). For BaM-UDA, we set $Q = 0.9$ and use a one-minus-cosine warm-up scheduler of 50 epochs to the KL loss coefficient (resulting in a coefficient going from 0 to 1 in 50 epochs). For each benchmark, we swept the learning rate across $\{0.01, 0.001\}$ and select the best model for both the baseline and ours. The standard image augmentations used in vision problems cannot be applied to this problem, since it destroys the scientific integrity of the data (e.g. cropping the PhC input would result in a completely different band gap profile). Instead, we used the augmentations proposed in Loh et al. (2022) (periodic translations, rotations and mirrors)

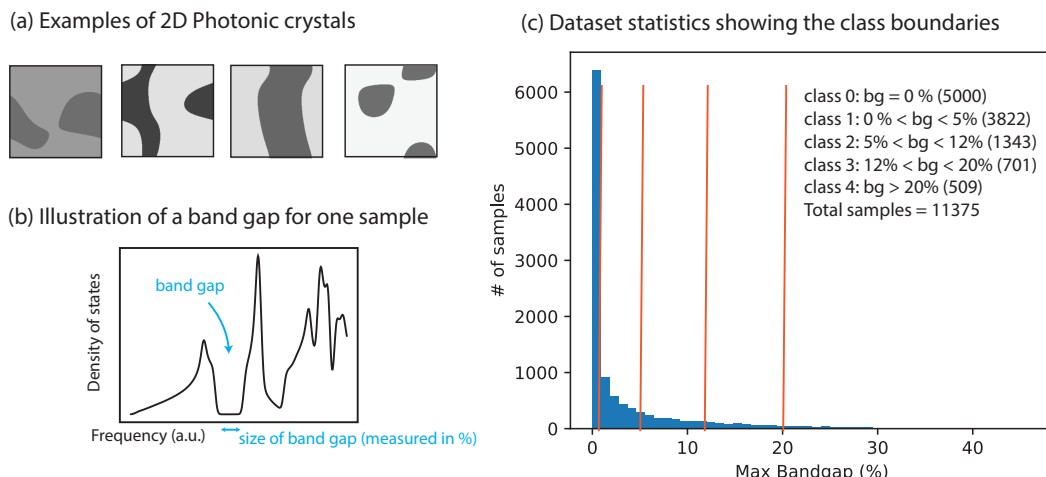

Figure 5: **Photonics dataset.** (a) Examples of 2D periodic photonic crystals (PhC), (b) illustration of the band gap size of a single PhC seen from its density-of-states, a common spectrum of interest for PhCs (Loh et al., 2022), and (c) dataset statistics for the task used in this work where the samples were binned into 5 classes based on their band gap (bg) with class boundaries detailed in the inset. Numbers in parenthesis show the number of samples in each class, showing strong class imbalance.

and applied them symmetrically (i.e. there is no distinction of strong and weak augmentations). In addition, we included these augmentations to the labeled samples as well.

## G   Ablation Studies

### G.1   Further ablation studies on BaM-

**Number of weight samples.**   Bayesian model averaging produces better calibrated predictions through "Bayesian marginalization", i.e. by averaging over multiple predictions instead of using a single prediction of the model. In Table 10, we show ablation studies on the number of weight samples taken from the variational posterior in the BNN layer used for "Bayesian marginalization", i.e. for computing the posterior predictive. We observe an overall trend that increasing the number of weight samples lead to better calibration and final test accuracies, providing direct evidence for the effectiveness in using Bayesian model averaging towards improving calibration. In order to limit the computational overhead arising from taking a large number of weight samples, given that pseudo-labeling happens at every iteration, we limit $M$ to 50 in our study. The computational overhead for $M = 50$ is discussed in Appendix H, where we see that incorporating Bayesian model averaging incurs negligible computational overhead.

Table 10: Ablation of number of weight samples, $M$, taken from the variational posterior in BaM-UDA. Dataset is CIFAR-100 with 400 labels. Highlighted in cyan is the main configuration used.

| $M$ | Test Accuracy | ECE |
|-----|---------------|-------|
| 2   | 50.0          | 0.403 |
| 5   | 58.1          | 0.342 |
| 10  | 58.5          | 0.336 |
| 50  | 59.7          | 0.327 |

**A basic semi-supervised learning setup.**   Many SOTA semi-supervised learning methods incorporate several techniques like weak and strong augmentation, sharpening of pseudo-labels and selection criteria to

achieve the best performance. An interesting study is to investigate a basic semi-supervised learning setup where many of these techniques are removed. In Table 5 of the main text, we investigated the importance of the different features in BaM- and found that even without a selection metric, BaM still leads to some performance gains. Here, we study a simple set up of semi-supervised learning, where we remove sharpening and the selection metric and compare this basic baseline with one where the last layer is replaced with a BNN via BaM-; results are shown in Table 11. Results show that performance gains via BaM- are consistent and are effective independent from the techniques commonly introduced in SSL methods.

Table 11: Basic semi-supervised learning setup without selection criteria.

|                    | CIFAR-100-400 | CIFAR-100-4000 |
| ------------------ | ------------- | -------------- |
| Basic setup        | 53.5          | 74.0           |
| Basic setup + BaM  | 54.6          | 74.9           |

## H   Computational Requirements and Additional Computational Costs

All CIFAR-10 and CIFAR-100 experiments in this work were computed using a single Nvidia V100 GPU. ImageNet experiments were computed using 64 Nvidia V100 GPUs. A key aspect of our proposed calibration methods is the requirement of adding minimal computational cost to the baseline approaches. In the following paragraphs we list the additional computational cost (based on wall-clock time on the same hardware) for our proposed methods.

**Computational cost of Bayesian model averaging.**   The main additional computational cost comes from executing several (in our case, 50) forward passes through the weight samples of the BNN layer when deriving predictions for the unlabeled samples. This additional overhead is minimal since only the final layer, which consists of a small fraction of the total network weights, is (approximate) Bayesian. On the benchmarks of CIFAR-100 where gains are larger, we found the BNN versions to take only around $2-5\%$ longer in wall-clock time on the same hardware when compared to the baseline, for the same total number of iterations. This justifies the BNN layer as a plug-in calibration approach with low computational overhead. The recorded run times for 500 epochs for each method on the same hardware are shown in Table 12.

Table 12: **Wall-clock run time (in hours) for 500 epochs for each method.**

|         | CIFAR-10 | | CIFAR-100 | | |
| ------- | --------- | ------------ | ---------- | ----------- | ------------ |
|         | 250 labels | 2500 labels | 400 labels | 4000 labels | 10000 labels |
| FM      | 30.8      | 19.3         | 77.2       | 74.7        | 71.9         |
| BaM-FM  | 36.5      | 21.5         | 79.7       | 75.1        | 73.7         |
| UDA     | 40.8      | 23.8         | 77.8       | 73.7        | 73.2         |
| BaM-UDA | 50.1      | 25.6         | 80.6       | 75.4        | 74.5         |

**Computational cost of SWA & EMA.**   Both SWA and EMA requires storing another network of the same architecture, denoted $g$ in the main text, which maintains the aggregate (exponentially weighted aggregate) of past network weights for SWA (EMA). Rather than using representations from the original backbone, $f$, representations from $g$ are used to derive the better calibrated predictions, and thus the main computational overhead comes from the second forward pass needed per iteration. We timed the methods and found that PAWS+SWA and PAWS+EMA took around 18% longer in wall clock training time (on the same hardware) when compared to the baseline PAWS method for each epoch of training. However, as shown in the main text, PAWS+SWA and PAW+EMA resulted in a significant speed up in convergence, cutting training time by $>60\%$ and additionally giving better test performances. Hence rather than an overhead, improved calibration in PAWS+SWA and PAWS+EMA in fact reduces the computation cost needed for the original approach.

## I Extreme low-label settings in CIFAR-100

Table 13 compares BaM-UDA against FixMatch and UDA in the extremely low-label settings for CIFAR-100 such as 100 labels and 200 labels. These are settings with extremely small number of labels since there is only 1 label per class (the lowest possible) for the 100 labels setting and 2 labels per class for the 200 labels setting. In the extreme setting of 1 label per class, BaM-UDA gives a decent 35.2% accuracy, a whopping 27% improvement from the UDA baseline (without the last bayesian layer) and a 19% improvement from the next best method (FixMatch).

Table 13: **Extremely low labels in CIFAR-100.**

|  | CIFAR-100 | | |
|---|---|---|---|
|  | 100 labels | 200 labels | 400 labels |
| FM | $16.0_{\pm 3.3}$ | $45.6_{\pm 0.3}$ | $56.4_{\pm 1.6}$ |
| UDA | $8.2_{\pm 0.7}$ | $19.0_{\pm 0.1}$ | $44.1_{\pm 0.7}$ |
| BaM-UDA | $\mathbf{35.2}_{\pm 1.3}$ | $\mathbf{51.2}_{\pm 0.2}$ | $\mathbf{60.3}_{\pm 0.6}$ |

## J Societal Impact and Ethical Considerations

Semi-supervised learning is arguably one of the most important deep learning applications, as in real life we often have access to an abundance of unlabeled examples, and only a few labeled datapoints. Our work highlights the importance of calibration in semi-supervised learning methods which could yield benefits for real-world applications in two main ways: 1) a better deep learning model that is more data efficient and 2) a model that is better calibrated and can quantify uncertainty better. The latter is highly important for crucial societal applications such as in healthcare. However, as with any deep learning application, there may be biases accumulated during dataset collection or assimilated during the training process. This issue may be more acute in a semi-supervised setting where the small fraction of labels may highly misrepresent the ground truth data. This may lead to unfair and unjust model predictions and give rise to ethical concerns when used for societal applications.

