# OpenReview forum: "Mitigating Confirmation Bias in Semi-supervised Learning via Efficient Bayesian Model Averaging"
_TMLR — Accepted by TMLR_

### Review · Reviewer_nAoU · 2023-05-02

**Summary Of Contributions:**

This paper replaces the last layer of a regular NN with a Bayesian one to compute calibrated uncertainty estimates for the data selection in semi-supervised learning. The techniques are simple, and the results are thorough. The paper also explores a different family of SSL methods that do not rely on a selection metric and makes some algorithmic modifications.

**Audience:**

Yes

**Claims And Evidence:**

Yes

**Requested Changes:**

Please address the concerns detailed in Weaknesses and Minor issues.

**Strengths And Weaknesses:**

Strengths
- The proposed method is simple to understand and implement.
- The experimental results cover multiple datasets such as CIFAR-10. CIFAR-100, and ImageNet.

Weaknesses
- In my opinion, the problem of semi-supervised learning in a low-shot regime is not significant. For example, the recent low-shot learning methods like CLIP-adapter and CaFo are much easier to use and can provide better performance. On the other hand, the recent fashion of zero-shot transfer enabled by large pre-trained models can already yield competitive enough results. Namely, I don't think the "16% improvement in test accuracy on the CIFAR-100 with 400 labels benchmark" is promising.

- The technique novelty is limited. A last-layer Bayesian treatment is introduced, stochastic VI is used for learning, and the uncertainty estimates are leveraged for data selection. The community has thoroughly investigated every part of this. The only novelty arises from that the selected data is used back for model training. But, I want to point out that this paper makes a close analogy with the Bayesian active learning pipeline developed by Yarin Gal. The difference is that you use the current model to label the selected data, while Gal uses an additional expert for data labeling.

- After reading the introduction, I thought the authors explored some theoretical connections between the confirmation bias in SSL and a Bayesian treatment. But, in fact, the findings are purely empirical. A deeper theoretical understanding is required to make the paper more convincing.

- For Section 7, I don't agree that SWA and EMA are approximate Bayesian inference or Bayesian model averaging approaches. You should make clear the difference between the weight-space average and the function-space average. So, I think including this section under the current paper titled ".. Efficient Bayesian Model Averaging" is improper. You can try SWAG, though I still don't regard it as a principled Bayesian inference way.

- The performance gain is not substantial. In particular, I really want to see a significant gain on the imbalanced CIFAR-10-LT.

Minor issues
- The $\rho_t$ is not an argmax operation but a one-hot operator at $t=0$. Right?

- "Bm contains the set of samples with ρ0(q) ∈ (m−1/M ,m/M ]". ρ0(q) is a model prediction or model confidence or a one-hot vector? The notations are confusing.

- In section 4, you should clarify how the other weights of the model are trained. They remain point estimates, so MLE or MAP is applied? You can find a unified interpretation in related BNN papers.

- "This constitutes a more accurate measure of model uncertainty compared to the maximum logit value". Is there a citation explaining why it is more accurate? Besides, other uncertainty estimates, such as mutual informantion and softmax variance, also apply here. You can empirically evaluate their effectiveness.

- Is the method hyperparameter-sensitive? What hyperparameters should be carefully tuned in a new task? Is there an ablation regarding this?

---

> ### Author Response · Authors · 2023-05-20
> **Response from authors (part 1)**
>
> We appreciate the reviewer’s time and thoughtful comments/questions. Individual points raised by the reviewer are addressed below.
>
> > In my opinion, the problem of semi-supervised learning in a low-shot regime is not significant. For example, the recent low-shot learning methods like CLIP-adapter and CaFo are much easier to use and can provide better performance. On the other hand, the recent fashion of zero-shot transfer enabled by large pre-trained models can already yield competitive enough results. Namely, I don't think the "16% improvement in test accuracy on the CIFAR-100 with 400 labels benchmark" is promising.
>
> We thank the reviewer for providing this perspective. While self-supervised learning has indeed shown to be highly effective, PAWS paper [1] has shown that semi-supervised learning methods still outperform self-supervised learning methods like BYOL, SimCLRv2, etc. With regards to the specific models that the reviewer mentioned, they require specialized image-text paired dataset which may not be readily available for specialized domains such as science and medical domains. For example, in our work, we also showed one such application of SSL to a problem in photonic science, where we showed that our method can outperform SOTA semi-supervised learning methods on the problem of bandgap prediction in photonic materials. Such domains are less amenable to self-supervised learning approaches since data collection is highly costly due to the need for laborious experiments or computationally expensive tools and thus semi-supervised learning still provides significant benefits over traditional supervised learning.
>
> [1] Semi-Supervised Learning of Visual Features by Non-Parametrically Predicting View Assignments with Support Samples. Assran et. al. (2021). (https://arxiv.org/pdf/2104.13963.pdf)
>
> > The technique novelty is limited. A last-layer Bayesian treatment is introduced, stochastic VI is used for learning, and the uncertainty estimates are leveraged for data selection. The community has thoroughly investigated every part of this. The only novelty arises from that the selected data is used back for model training. But, I want to point out that this paper makes a close analogy with the Bayesian active learning pipeline developed by Yarin Gal. The difference is that you use the current model to label the selected data, while Gal uses an additional expert for data labeling.
>
> We thank the reviewer for the ideas. The analogy with Bayesian active learning was also pointed out by reviewer v2un. We have now added some discussion in the related works section to highlight the resemblance of our approach to those used for sample selection in Bayesian active learning.
>
>
> Indeed, as the reviewer has pointed out, stochastic variational inference (SVI) has been used extensively for uncertainty estimation; however, to the best of our knowledge, we have not seen existing works applying them in semi-supervised learning and in particular, to address the issue of confirmation bias. Our work aims to highlight the systematic lack of consideration of uncertainty estimation in semi-supervised learning and to provide evidence that simple uncertainty estimation tools like SVI can be used to effectively mitigate confirmation bias.
>
> > After reading the introduction, I thought the authors explored some theoretical connections between the confirmation bias in SSL and a Bayesian treatment. But, in fact, the findings are purely empirical. A deeper theoretical understanding is required to make the paper more convincing.
>
> In Appendix A, we explored some theoretical connections of BaM- to generalization bounds. Despite this, we do not make any claims about proving theoretical connections between confirmation bias and Bayesian approaches, as we also state in our contributions that results are empirically demonstrated. Our goal is to use experiments to provide thorough evidence to support our claims that simple uncertainty estimation tools like SVI, when incorporated in SOTA SSL methods can help mitigate confirmation bias. We believe that this contribution is aligned with TMLR’s charter.

---

> ### Author Response · Authors · 2023-05-20
> **Response from authors (part 2)**
>
> > For Section 7, I don't agree that SWA and EMA are approximate Bayesian inference or Bayesian model averaging approaches. You should make clear the difference between the weight-space average and the function-space average. So, I think including this section under the current paper titled ".. Efficient Bayesian Model Averaging" is improper. You can try SWAG, though I still don't regard it as a principled Bayesian inference way.
>
> We thank the reviewer for this comment. We have included further clarifications to make it explicit that SWA and EMA perform weight-space averaging while typical BMA approaches (including BaM) perform function-space averaging.
> We agree that it may be less appropriate to classify them under BMA as well and have thus moved this entire section on SWA and EMA to the appendix, since the other reviewers also felt that the addition of this section was confusing.
>
> > The performance gain is not substantial. In particular, I really want to see a significant gain on the imbalanced CIFAR-10-LT.
>
> The magnitude of gains for each benchmark is inherently linked to the seriousness of confirmation bias, since BaM was designed to tackle confirmation bias using better uncertainty estimates. Simpler benchmarks like CIFAR-10 and low imbalance CIFAR-10-LT often do not suffer from serious confirmation bias and hence gains from improving uncertainty estimates is expected to be less. One way to visualize confirmation bias is to look at the selection purity, which shows the oracle accuracy of the unlabeled samples that were accepted by the model, like in Figure 2(c) of the main text. For the baseline method for CIFAR-10-LT at $\alpha=10$, we found the maximum purity rate to be about 96%, thus indicating that the model was mostly already correct whenever it accepted a sample and thus the issue of confirmation bias is mild and we should not expect huge improvements from incorporating BaM. (In contrast, on a benchmark where confirmation bias is serious, e.g. max. purity rate of 42% for the baseline of CIFAR-100-400, BaM achieves a more significant 16% performance gain.)
>
> Additionally, we would like to highlight that while performance gains on a single benchmark may not seem substantial, we compared BaM against a series of benchmarks and BaM consistently led to performance gains over the baseline across all of them (even up to 16% improvement on CIFAR-100). While the reviewer may consider even a 16% improvement to be not substantial, we feel such consistency in performance gains across all benchmarks to be strong evidence towards our claims that BaM effectively mitigates confirmation bias in SSL.
>
> ### Minor issues
> > The $\rho_t$ is not an argmax operation but a one-hot operator at $t=0$. Right?
>
> We thank the reviewer for pointing this out. $\rho_t$ is indeed a one-hot operation instead of an argmax, so we have corrected the manuscript and included further clarifications.
>
> > "Bm contains the set of samples with $\rho_0(q) \in (m−1/M ,m/M ]$. $\rho_0(q)$ is a model prediction or model confidence or a one-hot vector? The notations are confusing.
>
> We have corrected the notation to explicitly define that we are considering the model’s confidence, or the model prediction at the most likely class prediction. We apologize for the confusion.
>
> > In section 4, you should clarify how the other weights of the model are trained. They remain point estimates, so MLE or MAP is applied? You can find a unified interpretation in related BNN papers.
>
> We have added clarifications to section 4 to explain that the rest of the non-bayesian weights of the network remain as point estimates and are trained with MLE.

---

> ### Author Response · Authors · 2023-05-20
> **Response from authors (part 3)**
>
> ### Minor issues (continued)
> > "This constitutes a more accurate measure of model uncertainty compared to the maximum logit value". Is there a citation explaining why it is more accurate? Besides, other uncertainty estimates, such as mutual informantion and softmax variance, also apply here. You can empirically evaluate their effectiveness.
>
> We empirically evaluated their effectiveness from ablation studies in the newly added Table 5 of the main text, which shows that using the variance of predictions as a measure of model uncertainty is more effective than the maximum logit value. We reproduce the results here for convenience;
> |                                                           | CIFAR-100-400_labels | CIFAR-100-4000_labels |
> |-----------------------------------------------------------|----------------------|-----------------------|
> | BaM- using the max logit value as uncertainty measure     | 54.2                 | 74.5                  |
> | BaM- using variance of predictions as uncertainty measure | 59.7                 | 75.3                  |
>
> The initial goal of that sentence was to highlight that the maximum logit value has a poor interpretation of uncertainty, as shown in many calibration papers such as [2]. To avoid confusion, we have improved the wording of that line and further clarified that our approach is empirically verified to be more effective.
>
> [2] On calibration of modern neural networks, Guo et. al. (https://arxiv.org/abs/1706.04599)
>
> > Is the method hyperparameter-sensitive? What hyperparameters should be carefully tuned in a new task? Is there an ablation regarding this?
>
> Since BaM only modifies the last layer and retains most of the backbone of the original implementation, we used all the hyperparameters that were recommended by the original implementations. The only new hyperparameters we introduce are those coming from BaM’s formulation. We have included ablation studies on the key hyperparameters in the newly added section 7 and in Appendix G.

---

### Review · Reviewer_pZ95 · 2023-05-03

**Summary Of Contributions:**

The high level idea appears to be using a last-layer bayesian neural network (trained with elbo) to give better calibrated and better uncertainty aware pseudo labels for semi supervised learning.

However, the paper is a bit of a grab bag of a lot of little pieces.  First, there is a selection criteria used based on the variance of the posterior predictive to decide whether an example is worth training on or not, also most of the results in the paper are obtained by combining the last layer bayesian model averaging with Unsupervised Data Augmentation (UDA), and a strong and weak augmentation scheme, along with a sharpening operation applied to the posterior.

Later in Section 7, weight averaging is used with PAWS for another set of experiments.

**Audience:**

Yes

**Broader Impact Concerns:**

None.

**Claims And Evidence:**

No

**Requested Changes:**

How does BAM by itself perform at SSL?  I would have liked to see ablations without the acceptance criteria applied, merely using the soft pseudo-labels from the posterior predictive as the target.  If the posterior predictive has higher variance, that will already effect the KL loss if the soft labels are used, I don't see why you would also need a hard selection criteria, that seems to be at odds with the notion that the posterior predictive is well calibrated.

**Strengths And Weaknesses:**

Overall, the paper is very difficult to work though.  There is a lot going on and a lot of moving pieces.

The introduction of the paper focuses on the ability of bayesian models and the marginalization inherent in the posterior predictive as affording better calibration and uncertainty quantification which ought to be useful for SSL.  Why not focus on that?  I want to know whether the bayesian model averaging aids SSL, but given all of the moving pieces here I am not convinced by the evidence provided that it was BAM and any calibration it provided that provided the benefits.

I feel as though the simplest baseline and first direct test of the efficacy of BAM would be to start with a very basic UDA setup, where you simply take unlabeled data and two augmentations for the data and try to match the predictive distributions.  No selection criteria, no sharpening, just minimize the KL between the predictive distributions for the two augmented inputs.

As it stands, even though spending significant time with the paper, I'm not convinced I fully understand all of the details of what was done, would not be able to replicate the experiments and so am unconvinced that the gains are for the reasons described.

---

> ### Author Response · Authors · 2023-05-20
> **Response from authors (part 1)**
>
> We appreciate the reviewer’s time and thoughtful comments/questions. Individual points raised by the reviewer are addressed below.
>
> > Overall, the paper is very difficult to work though. There is a lot going on and a lot of moving pieces.
> The introduction of the paper focuses on the ability of bayesian models and the marginalization inherent in the posterior predictive as affording better calibration and uncertainty quantification which ought to be useful for SSL. Why not focus on that? I want to know whether the bayesian model averaging aids SSL, but given all of the moving pieces here I am not convinced by the evidence provided that it was BAM and any calibration it provided that provided the benefits.
>
> We acknowledge that it could be confusing since many SSL methods often include a huge bag of techniques (i.e. the “moving pieces” mentioned) to achieve SOTA performance. However, we would like to clarify that many of these moving pieces mentioned by the reviewer (strong and weak augmentation scheme, sharpening operation and selection criteria) were not introduced by us but were already present in the original baseline methods of UDA and FixMatch. We initially did not remove them in order to convincingly justify that the incorporation of BaM leads to an improvement over these already established SOTA baselines. If we had removed some of the mentioned moving pieces from the baselines, it could cast doubt on whether the improvements obtained from BaM were instead coming from the removal of features that could have been crucial for the original baselines. However, we agree that these could be interesting ablations, which we have now added (see the next discussion point).
>
>
> We have included further clarifications in the main text and Figure 1 to clearly highlight the only 2 additional modifications/features that were introduced in BaM – namely 1) averaging over samples and 2) variance-based selection metric.
>
>
> We further acknowledge the reviewer’s concern that section 7 on PAWS could add significant confusion, which has also been echoed by reviewer v2un. We have restructured the paper to move these experiments to the appendix and instead replaced this section with more important ablations, that the reviewer had suggested below, as we thought they would provide further insight to BaM.
>
> > I feel as though the simplest baseline and first direct test of the efficacy of BAM would be to start with a very basic UDA setup, where you simply take unlabeled data and two augmentations for the data and try to match the predictive distributions. No selection criteria, no sharpening, just minimize the KL between the predictive distributions for the two augmented inputs.
>
> We thank the reviewer for this interesting suggestion. We have added experiments where we used a basic version of a SSL setup with the selection criteria and sharpening removed, as the reviewer suggested. More specifically, the main difference BaM does is to simply create a better calibrated posterior predictive via bayesian averaging. Results are in Appendix G and reproduced below for convenience;
>
> |                                              | CIFAR-100-400_labels | CIFAR-100-4000_labels |
> |----------------------------------------------|----------------------|-----------------------|
> | Basic setup (no selection, no sharpen)       | 53.5                 | 74.0                  |
> | Basic setup (no selection, no sharpen) + BaM | 54.6 (+1.1%)         | 74.9 (+0.9%)          |
>
> This set of experiments show that the gains from BaM are consistent and are independent from several of the “moving parts” of SSL methods (e.g. selection criteria, sharpening).
>
> > As it stands, even though spending significant time with the paper, I'm not convinced I fully understand all of the details of what was done, would not be able to replicate the experiments and so am unconvinced that the gains are for the reasons described.
>
> As we expanded upon earlier, BaM only introduces two modifications/features. In fact, as we show in our pseudocode in Algorithm 1, adding BaM into the current baselines only requires a few extra lines of code. We hope that the additional ablations we show in the point above and the point below are sufficient to convince the reviewer that BaM results in consistent gains. Upon acceptance, we will also release our code in a public repository to aid the reproduction of experiments.

---

> ### Author Response · Authors · 2023-05-20
> **Response from authors (part 2)**
>
> > How does BAM by itself perform at SSL? I would have liked to see ablations without the acceptance criteria applied, merely using the soft pseudo-labels from the posterior predictive as the target. If the posterior predictive has higher variance, that will already effect the KL loss if the soft labels are used, I don't see why you would also need a hard selection criteria, that seems to be at odds with the notion that the posterior predictive is well calibrated.
>
> We thank the reviewer for this suggestion. We point the reviewer to the above points for the result of this ablation. Based on this results, we see that it is still beneficial to have a selection criteria to work in conjunction with a well-calibrated posterior predictive;
> i.e. on CIFAR-100-400_labels, BaM-UDA no selection criteria achieves 54.6% test accuracy while BaM-UDA with variance selection criteria achieves 59.7%.
>
> In addition, we would like to point the reviewer to another set of ablations in Table 5 of our newly added section 7 which further ablates the effect of the selection criteria. For convenience, we reproduced the results here;
> |                                         | CIFAR-100-400_labels | CIFAR-100-4000_labels |
> |-----------------------------------------|----------------------|-----------------------|
> | UDA (i.e. logit-based selection)        | 44.0                 | 72.9                  |
> | BaM-UDA (with logit-based selection)    | 54.2                 | 74.5                  |
> | BaM-UDA (with variance-based selection) | 59.7                 | 75.3                  |
>
> As expanded earlier, BaM introduces two new features: 1) averaging over samples and 2) variance-based selection metric. From this ablation, we see that (1) alone gives significant improvement via a more calibrated posterior predictive and further adding (2) gives further gains. We agree with the reviewer that these ablations may provide further insight to explain the efficacy and BaM and thus have included some of these ablations to the main text in section 7.

---

### Review · Reviewer_v2un · 2023-05-07

**Summary Of Contributions:**

This paper presents a Bayesian treatment for reducing confirmation bias in semi-supervised learning (SSL). The main idea is to estimate the uncertainties of the predictions and use them to build a selection criterion for pseudo-labels. The paper uses the variational approximation to the posterior of BNN with only the last layer treated random, and shows that the variance in the predictions computed with multiple samples from the variational posterior could serve as a decent selection criterion for SSL. For the SSL methods without selection criteria, the paper proposes to use EMA or weight-averaged parameters for pseudo-labeling, and demonstrate that those methods outperform the proposed method.

**Audience:**

Yes

**Broader Impact Concerns:**

The authors addressed societal impact and ethical considerations in the appendix.

**Claims And Evidence:**

Yes

**Requested Changes:**

- While I agree with the generic idea of using better-calibrated predictions for SSL and building such calibrated predictions from BNNs, I'm a bit skeptical about the choice the authors made for the actual implementation. The proposed method (BaM) is based on the last-layer BNN with variational posterior. One may choose one of those options for scalability issues but why use both? For instance, if the variational approximation is employed, BNN with full stochasticity for all weights are still amenable (unless you are using extremely large neural networks), or if the last layer BNN is employed one may consider more accurate posterior approximations such as HMC. Since the quality of uncertainty estimation is crucial for the efficacy of SSL, it would be important to justify the choice made in the paper.

- I'm a bit confused to see the experiments since PAWS+SWA or PAWS+EMA outperform BaM+PAW and the authors actually admit that. What is the main contribution of the paper? Is it only the BaM (last layer BNN + variational approximation + variance-based selection criterion), or the generic concept of applying uncertainty-based predictions for SSL? It would be important to clarify this, and if the latter is the main contribution, the paper might need some rewriting.

- The philosophy of the proposed method, especially the part where the uncertainty of the prediction is measured through some sort of disagreement (variance) between the predictions computed from BNN posteriors. This is somewhat similar in the way the acquisition functions are computed in uncertainty-based active learning, to give an example, BALD or BatchBALD. It would be good to have a discussion in the related work part.

**Strengths And Weaknesses:**

- The paper is well-written and easy to follow.
- The paper is tackling an important problem in SSL, and the suggested solution is a natural way to go.
- The performance of the proposed method, compared to the baselines with the selection criteria, is good.

---

> ### Author Response · Authors · 2023-05-20
> **Response from authors (part 1)**
>
> We appreciate the reviewer’s time and thoughtful comments/questions. Individual points raised by the reviewer are addressed below.
>
> > While I agree with the generic idea of using better-calibrated predictions for SSL and building such calibrated predictions from BNNs, I'm a bit skeptical about the choice the authors made for the actual implementation. The proposed method (BaM) is based on the last-layer BNN with variational posterior. One may choose one of those options for scalability issues but why use both? For instance, if the variational approximation is employed, BNN with full stochasticity for all weights are still amenable (unless you are using extremely large neural networks), or if the last layer BNN is employed one may consider more accurate posterior approximations such as HMC. Since the quality of uncertainty estimation is crucial for the efficacy of SSL, it would be important to justify the choice made in the paper.
>
> We thank the reviewer for these ideas. Our choice of using a last-layer BNN with variational posterior was motivated by a balance between computation and uncertainty estimation efficacy.
> While it may, in principle, be possible to use HMC instead of stochastic variational inference (SVI) to train the last layer BNN, to the best of our knowledge, training such a network could be highly intractable. We would have to run the HMC sampler to completion after every single backward pass/iteration of the non-Bayesian backbone and given that these SSL methods typically take around $10^6$ steps to converge, this could lead to an enormous computation overhead. Further, just the last layer of the network also involves quite a large number of parameters, such as $512\times100$ (e.g. in CIFAR-100), making it expensive for HMC, even if we were to run HMC for a small number of steps. In contrast, we show that using SVI on a last-layer BNN only adds about 2-5% in wall clock time on the CIFAR-100 benchmarks, while giving a significant improvement in uncertainty estimation (as evident from the decrease in ECE) compared to the full-non-BNN baseline.
>
> Similarly, using SVI on the entire network as opposed to just the last layer could also have similar computational considerations, since the network is not particularly small, e.g. there are around 23M parameters for CIFAR-100. Further, our last layer approach was also motivated by the fact that full SVI often leads to poor overall model accuracy, as demonstrated in [1].
> As such, we proposed to use a last-layer BNN via SVI in order to strike a balance between model accuracy, uncertainty estimation efficacy and computational overhead. We thank the reviewer for these ideas and have added a new paragraph in section 4.1 to list some motivations and justifications to support our choice.
>
> [1] How Good is the Bayes Posterior in Deep Neural Networks Really? Wenzel et. al. (https://arxiv.org/pdf/2002.02405.pdf)
>
>
> > I'm a bit confused to see the experiments since PAWS+SWA or PAWS+EMA outperform BaM+PAW and the authors actually admit that. What is the main contribution of the paper? Is it only the BaM (last layer BNN + variational approximation + variance-based selection criterion), or the generic concept of applying uncertainty-based predictions for SSL? It would be important to clarify this, and if the latter is the main contribution, the paper might need some rewriting.
>
> The original purpose of the experiments on PAWS+SWA and PAWS+EMA were intended as a supplemental study to investigate if this issue of confirmation bias is generic across all SSL methods and if similar tools could be useful. We however agree with the reviewer that adding these experiments may lead to confusion and may derail readers from our main contribution of the proposed method, BaM, of which we had provided extensive experiments and evidence for. This confusion was also echoed by reviewer pZ95. We have thus moved the entire section 7 on “Exploring SSL methods without a selection metric”  into the appendix, and instead replaced this section with more ablation results on BaM, that could provide further insight towards the efficacy of BaM. We thank the reviewer for raising this point.
>
> The main goal of our work is to highlight that, given that confirmation bias is an innate issue in SSL, the notion of uncertainty estimation or calibration should be of paramount importance; yet these notions are often missing in SOTA SSL methods. We wanted to show that incorporating simple off-the-shelf tools targeted at uncertainty estimation can lead to improved SSL performance.

---

> ### Author Response · Authors · 2023-05-20
> **Response from authors (part 2)**
>
> > The philosophy of the proposed method, especially the part where the uncertainty of the prediction is measured through some sort of disagreement (variance) between the predictions computed from BNN posteriors. This is somewhat similar in the way the acquisition functions are computed in uncertainty-based active learning, to give an example, BALD or BatchBALD. It would be good to have a discussion in the related work part.
>
> We thank the reviewer for providing this interesting perspective. Indeed, our proposed method BaM bears some resemblance to the spirit behind these acquisition functions, with a key difference being that while BALD selects samples with highest disagreement, BaM selects samples with the highest agreement. We have added some discussion around this in the related work section.

---

### Decision · Action_Editors · 2023-06-17

**Recommendation:** Accept with minor revision

**Comment:**

The authors study popular semi-supervised learning (SSL) methods that produce pseudolabels during training, such UDA and FixMatch. They observe that the main difficulty here is that if the model is miscalibrated, low-confidence pseudolabels are included in training and so will reinforce downstream error. Their proposed solution is Bayesian model averaging, which is a standard way to improve calibration. This is done by inserting a Bayesian neural network at the last layer trained via variational inference. An additional innovation is to use a selection step to filter out high-variance pseudolabels.  The authors compare this two-step approach, obtaining small but consistent improvements over popular SSL methods in several datasets along with improved calibration.

There were a number of comments from reviewers; the authors provided a substantial response to all of these points and made a fairly solid revision that improves the clarity of the work.

Overall I believe the paper meets the bar. The basic principles introduced by the authors (ie, improving uncertainty quantification is crucial for pseudolabeling-based SSL methods) are both sound and useful (as also noted by several of the reviewers). The proposed implementation is a reasonable way to achieve these.

Some concerns raised by the reviewers include,
- The value of SSL as a whole given zero-shot models. The authors have a reasonable answer here---and in any case, the pseudolabels produced by SSL can always be useful for fine-tuning zero-shot models.
- Small improvements from chaining multiple tricks. This is almost inevitable in a fairly mature field like SSL, where a huge number of ideas have been proposed and combined over the last 20+ years.

For this reason, the paper is heading towards acceptance.

I recommend the following minor revisions. The paper, while making its basic points, could stand to be a bit more informative in some of its experiments, especially in two aspects: wall-clock time across experiments (rather than a few quoted ones), and the number of % of labeled VS unlabeled data. These two factors are important for the SSL audience. I suggest
- Adding a table in the appendix that shows change in wall-clock time for all of the settings ,
- The authors mention that the very low-labeled setting case is too sensitive to initalization. Fair enough---what is the gain when the smallest proportion of labeled VS unlabeled data where reasonable conclusions can be drawn? This should be in more favorable to BaM.





**Audience:**

Yes. The semi-supervised learning area is classic in ML but has also had attention and breakthroughs in the last few years. The authors propose an approach that mitigates one of the big issues with the best current techniques, so it is of interest.

**Claims And Evidence:**

Yes, and in particular, post-review the authors have a compelling and well-evidenced set of claims.